Resource

# IARA: a complete and curated atlas of the biogenesis of spliceosome machinery during RNA splicing

Kelren S Rodrigues[1], Luiz P Petroski[1], Paulo H Utumi[1], Adriano Ferrasa[2], Roberto H Herai[1,3]

**Splicing is one of the most important post-transcriptional processing systems and is responsible for the generation of transcriptome diversity in all living eukaryotes. Splicing is regulated by the spliceosome machinery, which is responsible for each step of primary RNA processing. However, current molecules and stages involved in RNA splicing are still spread over different studies. Thus, a curated atlas of spliceosome-related molecules and all involved stages during RNA processing can provide all researchers with a reliable resource to better investigate this important mechanism. Here, we present IARA (website access: https://pucpr-bioinformatics.github.io/atlas/), an extensively curated and constantly updated catalog of molecules involved in spliceosome machinery. IARA has a map of the steps involved in the human splicing mechanism, and it allows a detailed overview of the molecules involved throughout the distinct steps of splicing.**

## Introduction

The human genome consists of ~25,000 protein-coding genes (Abdellah et al, 2004). In the cascade of protein formation, the DNA, which contains the genes, is usually transcribed by the RNA polymerase II complex into a single-stranded molecule, called pre-messenger RNA (pre-mRNA) (Crick, 1970; Sainsbury et al, 2015). The pre-mRNA is then processed by a series of essential biochemical steps for post-transcriptional regulation, such as 5′ capping, 3′ polyadenylation, and splicing, which ultimately produces the mature mRNA that is translated into proteins by the ribosomes (Singh et al, 2015).

The pre-mRNA molecule is formed by alternated regions called exons and introns, and the splicing mechanism, one of the main post-transcriptional regulation processes in all eukaryotic cells, catalyzes the primary transcript by the removal of introns to join exons to produce the canonical mature mRNA molecule (Chow et al, 1977; Shi, 2017). However, the processing of pre-mRNA transcripts can work in alternative ways through alternative splicing (AS), by keeping introns or removing exons in a different manner to form isoforms of the canonical mRNA molecule (Chen & Manley, 2009). Thus, the AS can produce functionally distinct proteins, being responsible for a large increase in the variability of the cellular transcriptome and proteome (Pan et al, 2008). The splicing mechanism involves several steps for the processing of primary RNA and is formed by a large and complex molecular machinery called the spliceosome, which is mostly composed of small nuclear ribonucleoproteins (snRNPs) (Shi, 2017). Defects in the splicing machinery can trigger the dysregulation of mRNA isoform formation and might contribute to diseases, including mental disorders (Tazi et al, 2009). In humans, for example, ~95% of the multiexon genes are subjected to AS (Pan et al, 2008). Moreover, the number of spliced transcripts in eukaryotic cells is widely variable between species, with few introns per genome for some species, such as in mammal genomes, to thousands for other species, such as in plant genomes (Roy & Gilbert, 2006).

A better understanding of the splicing mechanism, including the discovery and mapping of novel molecules that are part of spliceosome machinery, was highly improved by the recent technological advances, coupled with new techniques in the field of molecular biology. The RNA-sequencing approach, that uses next-generation sequencing, allows a detailed analysis at the RNA level that enables to characterize the content and splicing isoform pattern (exon and intron distribution), at the nucleotide level, of the transcripts, and the expression level for each gene (Pan et al, 2008; Wang et al, 2009). Likewise, the cross-linking immunoprecipitation (CLIP) technique combined with high-throughput sequencing (CLIP-seq or HITS-CLIP) allowed the identification of RNA-binding proteins, which is based on the irreversible cross-linking between proteins and RNA through ultraviolet light and immunoprecipitation, which enabled the discovery of several high-affinity binding sites in intronic regions, in addition to novel molecules as part of spliceosomal components (Ule et al, 2005; Hafner et al, 2021). Another technique that has gained momentum is cryoelectron microscopy (cryo-EM), which allows the visualization of biological macromolecule structures in a near-atomic resolution, and coupled with bioinformatics approaches, it allows the determination of

[1]Laboratory of Bioinformatics and Neurogenetics, Graduate Program in Health Sciences (PPGCS), School of Medicine and Life Sciences, Pontifícia Universidade Católica do Paraná, Curitiba, Brazil   [2]Informatics Department, Universidade Estadual de Ponta GrossaPonta Grossa, Brazil   [3]Research Division, Buko Kaesemodel Institute, Curitiba, Brazil

Correspondence: roberto.herai@pucpr.br

structures of the spliceosome, enhancing our understanding of the splicing machinery (Fernandez-Leiro & Scheres, 2016). However, all current findings on spliceosome-related snRNPs are fragmented throughout distinct studies, with several reported molecules requiring manual curation to ensure their correct involvement with a specific step of splicing. Thus, a general snapshot presenting all the stages involved in RNA processing is a challenging task to be accomplished.

In this work, we present the most recent and curated catalog of spliceosome-related molecules of humans. We performed a literature review to map the genes participating in regulating different steps of splicing. Next, we manually curated and classified the genes according to their involvement in splicing through distinct steps of transcript processing. We then created an updated online resource with an atlas of all spliceosome-related genes and a concise description of their role in the regulation of splicing. We also present details about the architecture and molecular organization of the spliceosome during its activation and catalytic activity in humans. We conclude our review by presenting some diseases to exemplify how

defects in the splicing mechanism can cause distinct phenotypes in human diseases.

# Results

## Splicing steps for transcript maturation

Pre-mRNA splicing is regulated by a dynamic ribonucleoprotein complex known as spliceosome (Wahl et al, 2009). This process occurs in two sequential transesterification steps allowing the pre-mRNA molecules to excise the introns and join the exons to generate a mature messenger mRNA. The introns have three conserved regions: the 5' splice site (5'ss) located near the 5' end of the intron, the 3' splice site (3'ss) near the 3' end of the intron, and an intermediate region called the branch site (BS) (Krämer et al, 1984; Black et al, 1985; Will & Lührmann, 2011). In the first step of splicing, known as branching, a 2' hydroxyl region of conserved adenosine, the BS, attacks a phosphate at the 5'ss and results in the release of 5' exon and the formation of an intermediate intron known as lariat (Fig 1A)

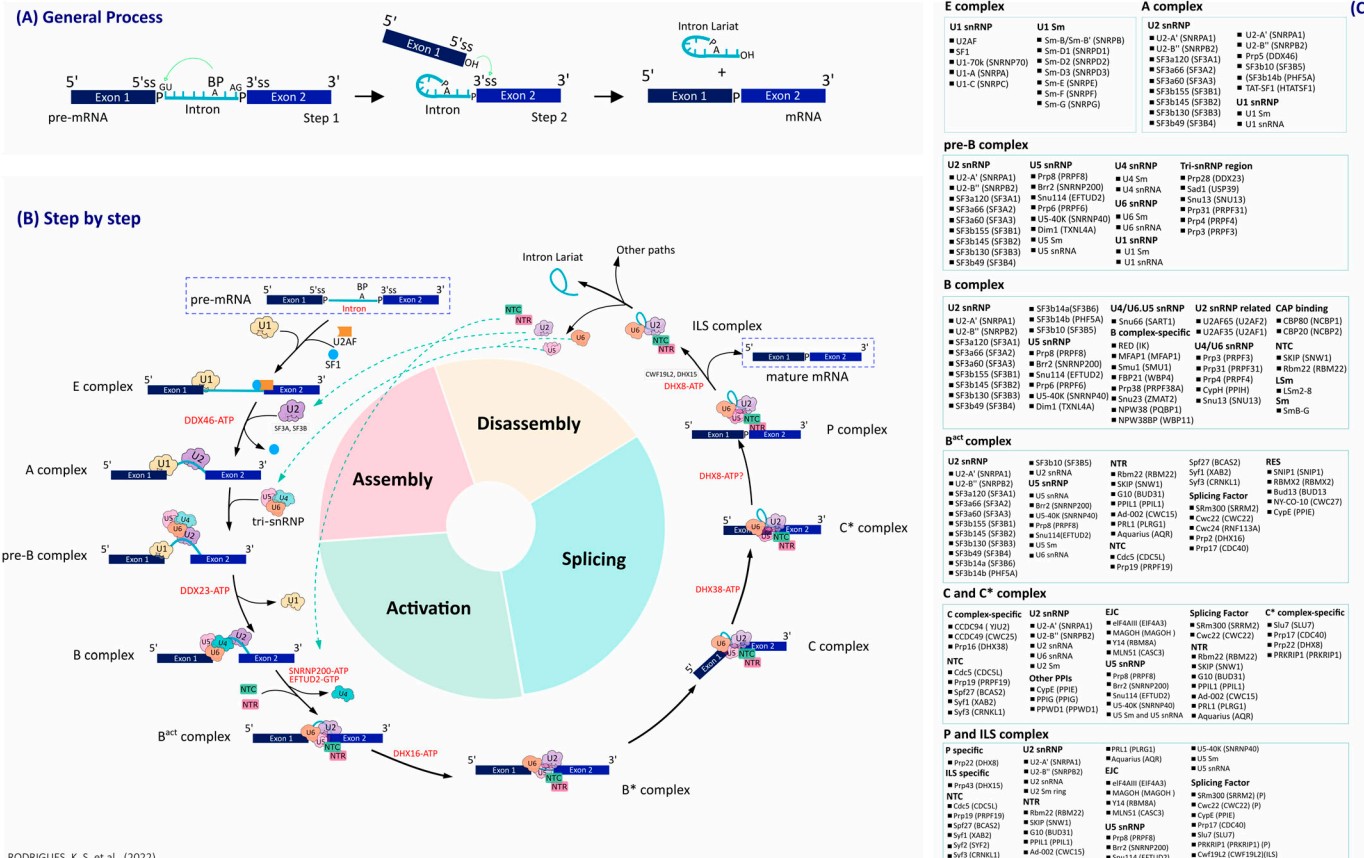

**Figure 1. Schematic overview of the complete cycle through all stages of the spliceosome machinery and corresponding molecules involved in each cycle.**
(A) In the first step of splicing, known as "branching," a 2' hydroxyl region of conserved adenosine (BS) of the pre-mRNA attacks a phosphate at the 5'ss and results in the release of exon 5' and formation of a lariat intermediate intron. The second step, called "exon ligation," allows the binding between the 5' and 3' exons of the transcript through the attack carried out by the hydroxyl group of the 5' exon to the 3'ss, which causes the release of the intron and binding of the exons, generating a mature mRNA. (B) 10 conformational states of the spliceosome are named as E, A, pre–B, B, B^act, B*, C, C*, P, and ILS complexes. Transitions between these states are regulated by helicases or ATPases and several other families of molecules. (C) List of molecules involved in spliceosome machinery and their classification into subgroups of complexes.

(Padgett et al, 1984; Wahl et al, 2009; Shi, 2017). The second step, called exon ligation, allows the binding between the 5′ and 3′ exons of the transcript through the attack carried out by the hydroxyl group of the 5′ exon to the 3′ss, which causes the complete release of the intron and binding of the exons (Wahl et al, 2009; Shi, 2017) (Fig 1A). The machinery involved in the splicing steps, the spliceosome, has five main molecules called small nuclear RNAs (snRNAs), the U1, U2, U4, U5, and U6, that are individually able to interact with several snRNPs, and with other splicing factors such as NTC (19 complexes) and NTR (19-related) complexes (Lerner & Argetsinger Steitz, 1979; Grabowski & Sharp, 1986; Chan et al, 2003; Shi, 2017).

Thus, for the complete processing of a primary RNA molecule, there are four key spliceosome-related steps: Assembly, Activation, Splicing, and Disassembly (Fig 1B). These key steps are subjected to several molecules, organized within different complexes, and are detailed in the following topics.

## E complex

During spliceosome assembly, the U1 snRNP complex interacts with a short sequence (nucleotides 3–10) at the 5′ intron splice site and stabilizes base pairing with the 5′ end of the U1 snRNA (Will et al, 1996; Kondo et al, 2015). In this step, U2AF and splicing factor 1 (SF1) bind to the 3′ splicing site and branch point site, forming the spliceosome E complex (Will et al, 1996; Kondo et al, 2015). The composition and spatial organization of the human U1 snRNP functional nucleus was determined through electron density maps, with crystal structures of resolution of 5.5, 4.4, 3.3, and 2.5 Å (Pomeranz Krummel et al, 2009; Weber et al, 2010; Kondo et al, 2015). These studies demonstrated the minimal structure of U1 snRNP with the 5′ splicing site RNA (Pomeranz Krummel et al, 2009); the U1 A70KF-RNA crystal structure of the SNRNP70 (U1-70k, residues 60–216) linked to the stem–loop I of U1 snRNA (Kondo et al, 2015); and the organizational structure of the seven proteins of the Sm complex in snRNAs (Weber et al, 2010).

During the formation of the E complex, the U1 snRNA molecule forms four stem–loops (SL1, SL2, SL3, and SL4) and the H helix, a region formed by the pairing between nucleotides 12–16 and 118–122 (Weber et al, 2010). The snRNP U1 has a minimum structure of seven Sm proteins (SNRPB [Sm-B/Sm-B′], SNRPD1 [Sm-D1], SNRPD2 [Sm-D2], SNRPD3 [Sm-D3], SNRPE [Sm-E], SNRPF [Sm-F], and SNRPG [Sm-G]), three specific U1 proteins (SNRNP70 [U1-70K], SNRPA [U1-A], and SNRPC [U1-C]), and the U1 snRNA (Pomeranz Krummel et al, 2009). The seven Sm proteins are present in all spliceosome U snRNPs, except for U6 (Achsel et al, 1999; Weber et al, 2010). In U1 snRNP, the Sm proteins form a heptameric ring around the nucleotides of the Sm site (a conserved region rich in uracil), which is placed between the SL3 and SL4 in the U1 snRNA, forming the central domain of the U1 snRNP complex (Weber et al, 2010). The spatial structure of the RNA helices is organized into a SL4 located at 3′ of the Sm site and four 5′ helices, with SL1 and SL2 coaxially stacked, and SL3 and H helix (Weber et al, 2010). The SNRNP70 and SNRPA proteins interact with SL1 and SL2 through their RNA-binding domain (Oubridge et al, 1994; Pomeranz Krummel et al, 2009; Weber et al, 2010). SNRPC stabilizes base pairing between the 5′ splice site of the pre-mRNA and the 5′ end of the U1 snRNA, forming hydrogen

bonds with the 2′ OH groups and with phosphate atoms of both nearby RNA strands at the splicing junction (Kondo et al, 2015; Bertram et al, 2017a).

The 5′ end of the U1 snRNA and the general structure of the U1 snRNP are stabilized by the N-terminal helices of SNRPD2 and SNRPB, in a particular orientation relative to the Sm ring (Pomeranz Krummel et al, 2009). SNRPD2 binds with the H helix and has conserved regions that enable N-terminal interaction with RNA and N-terminal SNRPB, which also interacts with RNA at base SL2 (Kondo et al, 2015; Pomeranz Krummel et al, 2009). The N-terminal region of the SNRNP70 protein may be involved in restricting the movement of SL1 in relation to the central domain (Pomeranz Krummel et al, 2009). The RBD of SNRNP70 surrounds the set of Sm proteins through a long α-helix, and its N-terminal portion helps the binding of the SNRPC protein to the central Sm domain by interacting with SNRPD3 (Kondo et al, 2015). SNRPC is crucial for E complex formation as it binds to the minor groove of the RNA duplex by pairing with the GU invariant portion of the 5′ junction site, and its zinc finger domain stabilizes the U1 RNA duplex snRNA and the 5′ junction site (Kondo et al, 2015; Pomeranz Krummel et al, 2009). The SNRPA protein binds to SL2 through its N-terminal RBD and positions it to interact with SNRPB and SNRPD1 in the Sm ring, making the RNA structure in the central domain even more stable (Kondo et al, 2015; Pomeranz Krummel et al, 2009).

## A complex

After U1 complex formation in the 5′ intron splice site, the A complex is assembled. At this point, the snRNA U2 interacts with the 5′ junction site (ss) and the BS of the intron (Krämer et al, 1984; Black et al, 1985; Wahl et al, 2009) (Fig 2). The molecular architecture of the human U2 complex was described using 3D cryoelectron microscopy (3D cryo-EM) and protein cross-linking data. Zhang et al (2020) determined the structure of the main subunit of A complex, the human 17S U2 snRNP (4.1 Å), exhibiting a bipartite organization of two domains, divided into a major U2 5′ module and a minor U2 3′ module (Zhang et al, 2020). The 5′ 17S U2 domain has the SF3B complex (SF3B3, PHF5A, SF3B5, and SF3B1$^{HEAT}$) as its main component is connected to the 3′ domain, which has the U2 Sm RNP core linked by U2-A′ and U2-B′′ to the U2 snRNA (Zhang et al, 2020). The SF3A complex (SF3A3 [SF3a60], SF3A2 [SF3a66], and SF3A1 [SF3a120]) is part of the 3′ domain and is involved in the bridge between the U2 Sm core and SF3B (Zhang et al, 2020). Subsequently, Cretu et al (2021) also presented a 3D cryo-EM structure (~3.1 Å central resolution) of the U2 5′ module and described the SF3B complex, two zinc finger domains of SF3A2 and SF3A3, and part of the substrate of intron paired with the U2 snRNA (Cretu et al, 2021).

Through an ATP-dependent reaction, the displacement of SF1 occurs, which enables the recruitment of U2 snRNP (Gao et al, 2008; Taggart et al, 2017). According to the proposed structures, U2 snRNP is essential for the accurate recognition of BS within the intron by the loop region called branch point–interacting stem–loop (BSL) in the U2 snRNA, forming the U2/intron duplex (van der Feltz & Hoskins, 2019; Zhang et al, 2020). The transition from BSL to the U2/intron duplex is promoted by the action of PRP5/DDX46, HTATSF1, and SF3A2 molecules (Perriman & Ares, 2010; Cretu et al, 2021). PRP5/DDX46 acts to stabilize the incorporation of U2 into the

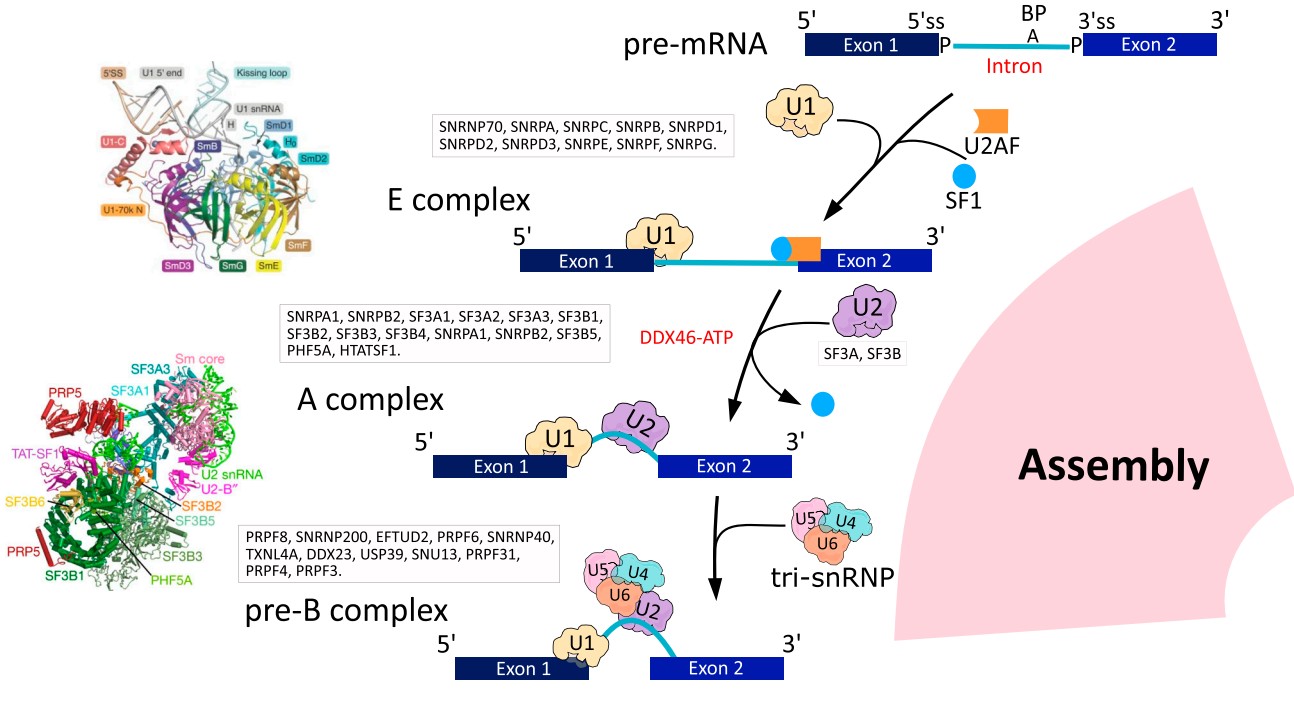

RODRIGUES, K. S. et al., (2022).

**Figure 2. During spliceosome assembly, the snRNPs U1 and U2 interact respectively with the 5′ splice site and the branching site of the intron, forming the spliceosome A complex.**
The initial and still unstable coupling of tri-snRNP (U4/U6.U5) to complex A forms the pre–B complex; at this time, U1 snRNA is still paired to the 5′ splice site, and the U2/U6 helix forms.

A complex and facilitate changes in the structure of U2 snRNP (Zhang et al, 2020). For this incorporation to occur, HTATSF1 is displaced from SF3B1 by the action of PRP5/DDX46, releasing BSL for the transition (Zhang et al, 2020; Cretu et al, 2021). It is also possible that in the BS recognition process, PRP5/DDX46 is acting on U2 snRNA remodeling (Zhang et al, 2020). After the release of BSL, SF3A2 binds to the precursor helix, facilitating its formation and stabilization. PRP5 covers SF3B1$^{HEAT}$ and assists in maintaining its conformation, which adopts an open state in 17S U2 snRNP, which becomes closed to stabilize the extended U2/intron helix near the 3′ss (A-to-B$^{act}$ spliceosome) (Bertram et al, 2017a; Cretu et al, 2021). The SF3B5, PHF5A, and SF3B3 subunits are similarly organized (Bertram et al, 2017a; Cretu et al, 2021).

In the following steps, the spliceosome goes through eight functional states that are divided into precursor of the pre-catalytic spliceosome (pre-B), pre-catalytic spliceosome (B), activated spliceosome (B$^{act}$), catalytically activated spliceosome (B*) (Agafonov et al, 2016; Bertram et al, 2017a; Zhang et al, 2018), catalytic step I complex (C), step II catalytically activated complex (C*) (Bertram et al, 2017b; Zhang et al, 2017; Zhan et al, 2018a), post-catalytic spliceosome (P), and intron lariat spliceosome (ILS) (Zhang et al, 2019).

### Pre–B complex

The initial and still unstable binding of tri-snRNP (U4/U6.U5) with U2 snRNP to complex A forms the pre–B complex. The U1 snRNA is still paired with the 5′ss, and the U5 and U6 snRNAs have not yet

recognized the pre-mRNA (Agafonov et al, 2016). After the association of the tri-snRNP, the transcript undergoes conformational changes that promote the formation of the U2/U6 duplex, the release of U1 mediated by the DDX23 helicase (Prp28), and the rearrangement of the SF3A and SF3B complexes of the tri-snRNP, generating the B complex (Bertram et al, 2017a; Zhan et al, 2018b) (Fig 3). The cryo-MS structure of the human pre–B complex (5.7 Å) demonstrated that the tetrahedron conformation of the tri-snRNP is composed of four vertices: the U4 Sm, U5 Sm, U6 LSm ring, and the SNRNP200 helicase (Brr2) (Zhan et al, 2018b). Between the tetrahedron edges of the U4 Sm and U6 LSm rings is the U2 snRNP, whereas the U1 snRNA is between the Sm-4 and Sm-5 rings (Zhan et al, 2018b). The pre–B spliceosome complex is the only one that has the five snRNPs present, formed by U1 and U2 snRNP, tri-snRNP (PRPF3 [Prp3], PRPF4 [Prp4], PRPF6 [Prp6], PRPF31 [Prp31], SNU13 [Snu13], TXNL4A [Dim1], and USP39 [Sad1]), U5 snRNP (PRPF8 [Prp8], EFTUD2 [Snu114], SNRNP200, SNRNP40 [U5-40k], U5 Sm, and U5 snRNA), U4 snRNP (U4 Sm and U4 snRNA), U4 snRNP (U6 Sm and U6 snRNA), and DDX23 (Zhan et al, 2018b).

The previously formed U2 intron/snRNA duplex is still present in the pre–B complex, and sequences from the 3′ end of the U6 snRNA to the 5′ end of the U2 snRNA form helix II (Anokhina et al, 2013; Zhan et al, 2018b). Also, U2 snRNP binds to tri-snRNP through interactions between the SF3B complex and the U6 LSm ring (Zhan et al, 2018b). The central nucleus shared between pre–B and B belongs to the U5 snRNP complex, having an almost invariable conformation, supported mainly by the N-terminal domain of PRPF8, together with EFTUD2, SNRNP40, the U5 Sm ring, and the U5 snRNA

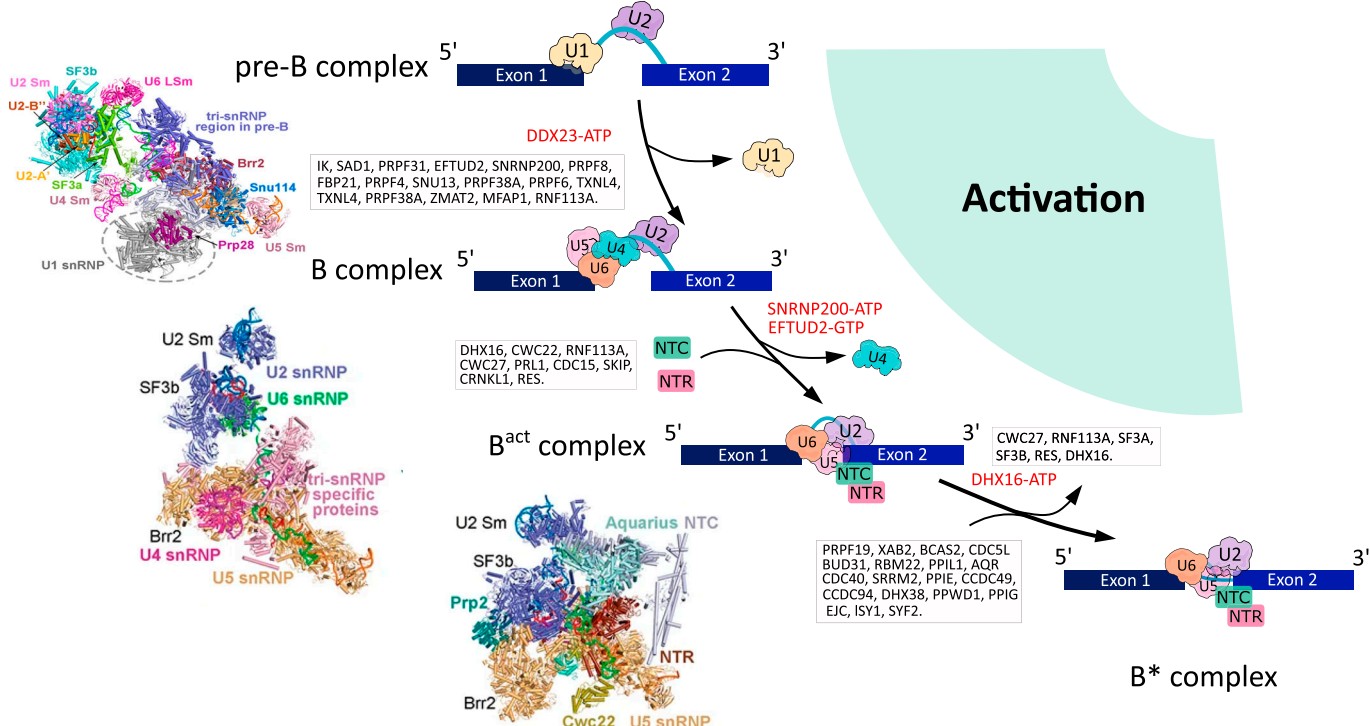

RODRIGUES, K. S. et al., (2022).

**Figure 3. In B complex, the transcript undergoes conformational changes, which is characterized by the formation of the stable binding of tri-snRNP with the transcript, the transcript positioning mediated by the DDX23 helicase of U6 in the 5′ss replacing U1, and the unwinding through SNRNP200 of the U4/U6 helices, which are extensively paired in the tri-snRNP complex, and with U4 dissociation, which triggers in a highly structured RNA network between the pre-mRNA and the snRNAs U2, U6, and U5, generating the spliceosome catalytic RNA core ($B^{act}$ complex).**
At this process, some proteins including the RES complex, NTC, and NTR proteins are recruited. By the activity and release of the DHX16, some splicing factors, SF3A, SF3B, and RES complexes, are dissociated. For the B* complex to occur, the vacant space likely allows the recruitment of DHX38 and the stage I–specific factors CWC25 and YJU2, the NTC proteins SFY2 and ISY1, the exon junction complex, and the PPIs PPWD1 and PPIG, allowing the branching reaction, generating a 5′ exon and an intron lariat-3′ exon intermediate (C complex).

molecule (Hang et al, 2015). The N-terminal domain of PRPF8 interacts with the helicase DDX23 (linked to U1 snRNP), whereas its C-terminal domain Jab1 links the tri-snRNP nucleus to the SNRNP200 protein (Mozaffari-Jovin et al, 2013; Nguyen et al, 2013). SNRNP200 also establishes this link through PRPF6 and USP39 (Hang et al, 2015). The PRPF6 protein forms two short α-helices on the exposed surface of TXNL4A that interacts directly with loop I of U5 snRNA (Zhan et al, 2018b). The transfer of base pairing between U1/5′ss snRNA and U6/5′ss snRNA starts the pre-B-to-B transition, which triggers further unwinding of the U4/U6 duplex by the SNRNP200 helicase (Zhan et al, 2018b). For this transition to occur, most tri-snRNP components undergo structural and organizational changes, including some PRPF8 domains (Zhan et al, 2018b). The dissociation of specific pre–B proteins (DDX23 and USP39) and the recruitment of new molecules to the B complex are also triggered (Zhan et al, 2018b).

## B complex

The pre-catalytic spliceosome stage is characterized by the following events: the formation of the stable binding of tri-snRNP with the transcript, the transcript positioning mediated by the DDX23 helicase of U6 in the 5′ss replacing U1, and then the unwinding through SNRNP200 (Brr2) of the U4/U6 helices, which are

extensively paired in the tri-snRNP complex, and with U4 dissociation, which triggers in a highly structured RNA network between the pre-mRNA and the snRNAs U2, U6, and U5, generating the spliceosome catalytic RNA core ($B^{act}$ complex) (Agafonov et al, 2016; Bertram et al, 2017a; Zhang et al, 2018) (Fig 3). Through the 3D cryo-EM structure (4.5 Å central resolution) of the human spliceosomal B complex, it described the molecular and spatial organization of several proteins (Bertram et al, 2017a). Within the stages involving the complexes A, B, and $B^{act}$, SF3A and SF3B proteins interact with pre-mRNA in the BS, acting to stabilize the U2/BS helix (Gozani et al, 1996; Bertram et al, 2017a; Zhang et al, 2018). The B complex–specific RED protein showed numerous cross-links with several snRNA U2 proteins and appears to play a role in bridging U2 with U5 proteins in the B complex (Ulrich et al, 2016; Bertram et al, 2017a; Zhang et al, 2021b). The USP39 (Sad1) protein plays a role in stabilizing the interaction of snRNPs U4/U6 and U5 and interacts with domains of PRPF31 (Prp31), PRPF8 (Prp8), EFTUD2 (Snu114), and SNRNP200, and may be involved in the function of keeping SNRNP200 in a pre-activation position, away from the duplex U4/U6 (Agafonov et al, 2016; Bertram et al, 2017a). This is consistent with the fact that SNRNP200 is present in the tri-snRNP complex and its activity is tightly regulated to ensure the correct unwinding of U4/U6 (Agafonov et al, 2016; Bertram et al, 2017a).

FBP21, a specific B complex protein, also helps to regulate the activity of SNRNP200, and interact with SNRNP200 and PRPF4 (Prp4) through its C-terminal domain (Bertram et al, 2017a; Henning et al, 2017; Kastner et al, 2019). The PRPF4 protein, in turn, interacts with several SF3A proteins, and with the helicase domain of SNRNP200 by multiple cross-linking (Bertram et al, 2017a; Kastner et al, 2019). Moreover, it has been reported that this protein helps in the binding of U4 and U6 complexes, interacts with regulators of SNU13, PRPF3 (Prp3), and PRPF6 (Prp6) splicing (contributing to the regulation of SNRNP200 and U4/U6 duplex unwinding), and phosphorylates PRPF6 and PRPF31, which may play an important role in tri-snRNP remodeling (Agafonov et al, 2016; Bertram et al, 2017a). Another protein, SMU1 (Smu1), can also help to stabilize the position of SNRNP200 after its rearrangement (Bertram et al, 2017a; Henning et al, 2017). The positioning of the U6 snRNA at the 5'ss is an essential step for the assembly and activation of the B* catalytic site. It has been proposed that the evolutionarily conserved TXNL4A (Dim1) protein plays a previously unknown direct role in 5'ss recognition in complex B (Agafonov et al, 2016; Bertram et al, 2017a). The PRPF38A (Prp38) protein interacts with the ZMAT2 (Snu23) and MFAP1 and is located close to the U6/5'ss helix, and seems to interact with an U6 motif, the ACAGA/5'ss sequence, and facilitates its repositioning during the activation phase of the splicing, in addition to helping to recruit proteins such as RNF113A (Cwc24) for the 5'ss to help in the transition of B into a B$^{act}$ complex (Bertram et al, 2017a; Henning et al, 2017; Schütze et al, 2016). Displacement of DDX23 was identified as a prerequisite for subsequent binding of the PRPF38A/ZMAT2/MFAP1 protein complex, whose binding site is near the U6/5'ss helix that is also mutually exclusive with that of DDX23 (Boesler et al, 2016; Bertram et al, 2017a).

## B$^{act}$ complex and B* complex

Another work reported the cryo-EM structures of the human B$^{act}$ complex (Zhang et al, 2018), which enables the mechanistic understanding of the steps involved in the formation of the B$^{act}$ complex and its transitions from complex B to complex B*. The authors of the study identified 52 protein components in the B$^{act}$ complex, with 11 belonging to U5 snRNP, 19 to U2 snRNP, 5 to NTC, 7 to NTR, 3 to the retention and splicing complex (RES) (SNIP1, BUD13, and RBMX2), three splicing factors (SRRM2 [SRm300], CWC22, and RNF113A), ATPase/helicase DHX16 (Prp2) and CDC40 (Prp17), and others (Fig 1C). As reported, the main constituents of U2 snRNP are the SF3A and SF3B complexes, and U2 includes all seven SF3B complex proteins (SF3B1, SF3B2, SF3B3, SF3B4, SF3B6, PHF5A, and SF3B5), three proteins from the SF3B5 complex (SF3A1, SF3A2, and SF3A3), U2 snRNA, and nine U2 snRNP core proteins that interact only with U2 snRNA (Fig 1C). The SF3B4 protein binds to SF3B2 outside of a superhelical structure made up of HEAT repeats and interacts with the upstream sequences of BPS (branch point sequence), stabilizing the U2/BPS duplex (Haselbach et al, 2018; Zhang et al, 2018). SF3B1 through a lateral opening of its superhelical structure is also linked to the U2/BPS duplex, and interestingly, SF3A2 is the only SF3A protein that specifically recognizes a RNA element in the U2/intron duplex (Will et al, 2002; Zhang et al, 2018). The RES complex interacts closely with the SF3B complex and consists of SNIP, RBMX2, and BUD13, playing an important role in pre-mRNA splicing and retention (Wysoczanski et al, 2014; Zhang et al, 2018).

Continuing the splicing, which is driven by the activity of the SNRNP200 helicase, the dissociation of U4 snRNP and U6 snRNP proteins from the B complex occurs, with several proteins including the recruitment of the RES complex (Agafonov et al, 2016; Bertram et al, 2017a; Zhang et al, 2018). Next, NTC and NTR proteins, together with NTD (N-terminal domain) from SF3A2, splicing factors SRRM2 and CDC40, and PPIE (CypE) are recruited, forming the mature B$^{act}$ complex (Haselbach et al, 2018; Zhang et al, 2018) (Fig 3). A late B$^{act}$ complex is formed by the release of the splicing factors RNF113A and CWC27, and it is further suggested that these steps may require pre-mRNA binding by DHX16 (Zhang et al, 2018). Despite a well-formed active site, the B$^{act}$ complex still cannot catalyze the branching reaction because of the spatial separation of BS from the 5'ss (Yan et al, 2016; Zhang et al, 2018). For the conversion of the B$^{act}$ complex to B* through the action of DHX16, the SF3A, SF3B, and RES complexes are dissociated, leading to the release of DHX16 (Zhang et al, 2018). The vacant space likely allows the recruitment of DHX38 (Prp16) and the stage I–specific factors CWC25 and YJU2, the NTC proteins SFY2 and ISY1, the exon junction complex, and the PPIs PPWD1 and PPIG, allowing the branching reaction to occur, generating a 5' exon and an intron lariat-3' exon intermediate (Zhang et al, 2018) (Fig 3).

## C complex

The branching reaction leads to the formation of the catalytic spliceosome of step I (C complex). Between complexes B* and C, there is no change in protein components. The cryo-EM structure of the human C complex demonstrated that the refined model of this complex contains 15,479 amino acids from 47 proteins, 414 nucleotides from three snRNAs (U2, U5, and U6), and pre-mRNAs (Zhan et al, 2018a). The 47 proteins in the atomic model include 11 from U5 snRNP, nine from U2 snRNP, seven from NTC, six from NTR complex, four from EJC (exon junction complex), five splicing factors (SRRM2, CWC22, CWC25, YJU2, and CDC40), four PPIs (designated as PPIL1, PPIE, PPIG, and PPWD1), and the DHX38 (Galej et al, 2016; Bertram et al, 2017b; Zhan et al, 2018a) (Fig 1C). The active site of human C complex comprises the intermolecular stem–loop of U6 snRNA, the catalytic triplex between U2 and U6, loop I of U5 snRNA, and five metal ions (Galej et al, 2016; Wan et al, 2016; Zhan et al, 2018a) (Fig 4). The conformations of the active-site RNA elements in the human C complex are supported by 15 surrounding protein components, particularly CWC25 and YJU2, the NTC component ISY1, the N-terminal domain of CDC5, CDC40, and the ribonuclease H–like domain (RNase H) of PRPF8 (Zhan et al, 2018a). The ATPase/helicases SNRNP200 and DHX38 in the peripheral regions are connected to the spliceosome nucleus mainly through CWC25 and YJU2, which interact with the 5' exon, the U2/BPS duplex, ISY1, and PRPF8 (Ohrt et al, 2013; Galej et al, 2016; Zhan et al, 2018a) (Fig 4). It is believed that DHX38 remodels the C complex by pulling the 3' end sequences of the intermediate intron lariat-3' exon (Galej et al, 2016; Zhan et al, 2018a) (Fig 4).

## C* complex

During the transition from C to C* (stage II catalytic spliceosome), the PRKRIP1 protein is recruited into the C* complex to stabilize the

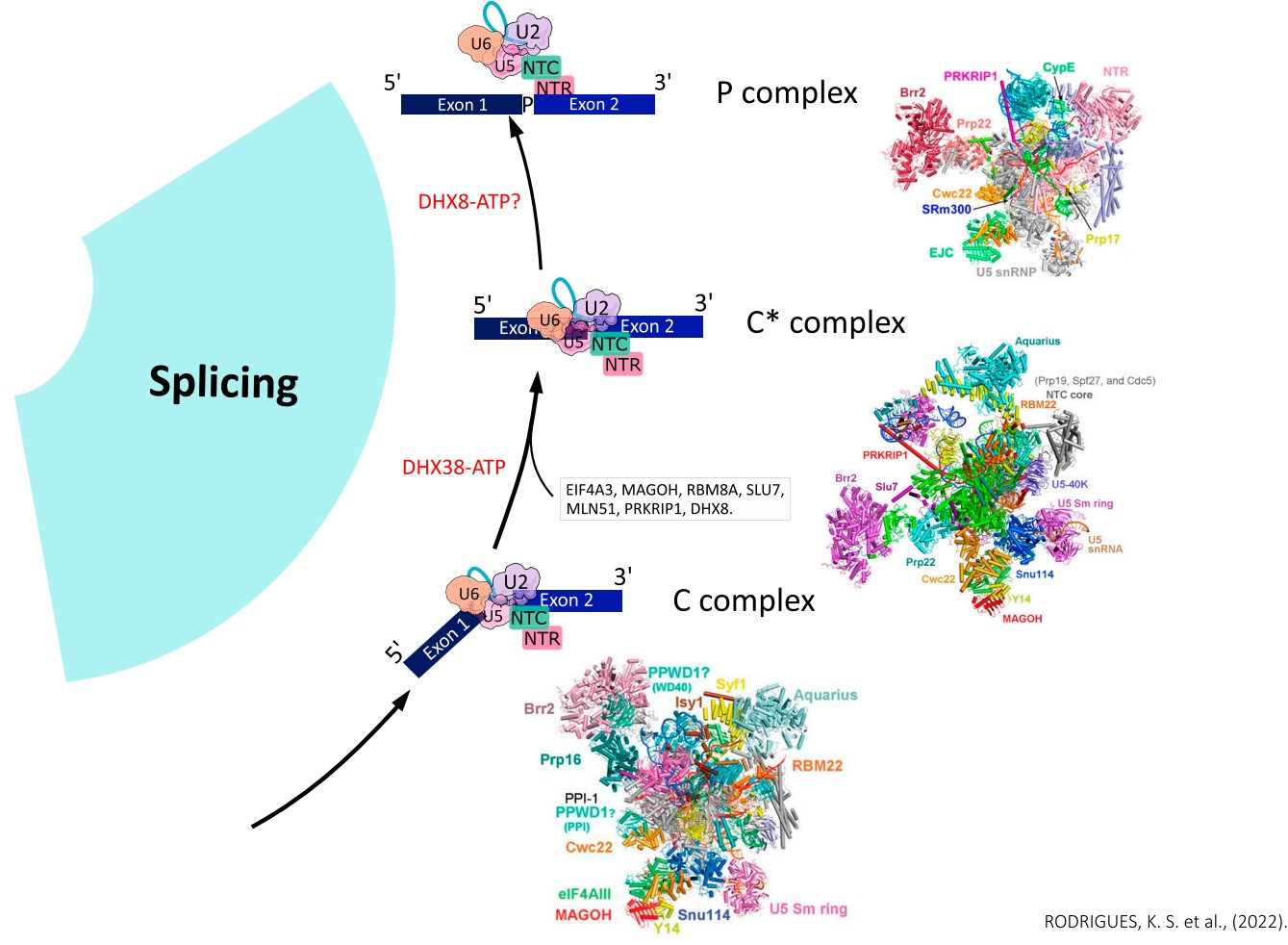

RODRIGUES, K. S. et al., (2022).

**Figure 4.   During the transition from C to C\*, the PRKRIP1 protein is recruited into the C\* complex to stabilize the new position of the U2 snRNP, SNRNP200 is translocated, DHX38 is dissociated in the transition allowing the binding of the DHX8 helicase, and second-stage factors SLU7, PRP18, and DHX8 are required to allow the juxtaposition of the 3′ OH of exon 5′ and the splice site 3′ (C\* complex), producing the ligated exons and the lariat intron (P complex).**

new position of the U2 snRNP, then the SNRNP200 is translocated and DHX38 is dissociated in the transition, and DHX8 (Prp22) is recruited into the C\* complex (Zhang et al, 2017; Zhan et al, 2018a) (Fig 4). Studies of the cryo-EM structure of the C\* complex (mean resolution of 3.76 and 5.9-Å) demonstrated a final atomic model of the human C\* complex, containing 14,496 residues of 46 proteins and 380 RNA nucleotides (Bertram et al, 2017b; Zhang et al, 2017). Through the enhanced resolution of 3.76 Å, the ATPase/helicase SNRNP200 proteins, the step II factor SLU7, the double-stranded RNA-binding protein PRKRIP1, and three EJC components (MAGOH, Y14, and MLN51) were identified with crucial roles in exon binding (Lerner & Argetsinger Steitz, 1979) (Figs 1C and 4). As described in catalytic step I, the cleaved 5′ exon remains attached to the spliceosome. After this step, the branched region of the intron must be displaced from the catalytic center of the spliceosome to allow the juxtaposition of the reagents from step II, the 3′ OH of exon 5′ (the nucleophile for step II) and the splice site 3′ (Bertram et al, 2017b; Zhang et al, 2017) (Fig 1A). For efficient positioning of the 3′ splice site at the catalytic center, the second-stage factors SLU7, PRP18, and DHX8 are required, consistent with the observation

that SLU7 is directly involved in the selection of the 3′ss (Chan et al, 2003) (Fig 4).

The SLU7 protein and the splicing factor CDC40 are placed on two opposite sides of the catalytic center (Zhang et al, 2017; Zhan et al, 2018a). SLU7 interacts mainly through different domains with PRPF8 and with the MA3 domain of the splicing factor CWC22, and it is located close to exon 5′ and probably, through the NTD, stabilizes its binding to the spliceosome, which facilitates the second stage of transesterification (Lerner & Argetsinger Steitz, 1979). On the contrary, CDC40 is closely associated with G10 and RBM22 (it makes direct contact with the intermolecular stem–loop of the U6 snRNA and binds to the intron lariat) and is above the extended duplex between the intron and U6 snRNA (Bertram et al, 2017b; Zhang et al, 2017). Furthermore, CDC40 forms a $\beta$-helix (residues 231–578), which, together with the associated molecules, is positioned between the BS/U2 duplex and the 5′ss/U6 duplex, stabilizing the conformation of the splicing active site (Lerner & Argetsinger Steitz, 1979) (Fig 4). At this stage, it was proposed that the RH domain of PRPF8 can act to stabilize the conformation of the branched structure of the intron, binding the nucleotides of the U2 snRNA and the U6 ACAGA

sequence (Grabowski & Sharp, 1986). It has been suggested that the RH domain can also help positioning the 3'ss to allow for the catalysis of step II (Grabowski & Sharp, 1986). DHX8 also directly interacts with PRPF8 and plays a role in positioning the 3' splice site for step II catalysis, and this is similar to what occurs after exon ligation, which is thought to bind and pull the 3' sequences of the ligated exon, releasing the exon of the P complex (Grabowski & Sharp, 1986; Chan et al, 2003) (Fig 4).

## P and ILS complexes

The final splicing step is composed of human P (exon bound remains bound) and ILS (exon is absent) complexes (mean resolutions of 3.0 and 2.9 Å, respectively), with ILS having two distinct conformations defined as ILS1 and ILS2 (Zhang et al, 2019). Through the analysis of the molecular mechanisms of these complexes, the processes of recognition of the 3'ss, exon release, and spliceosome disassembly were described (Fig 5).

The atomic model of the human P complex contains 44 proteins and four RNAs, totalizing 14,184 amino acids and 462 nucleotides (Zhang et al, 2019). The P complex, similar to the human C* complex, except for the changes in pre-mRNA, is the bond between the 5' exon and 3' exon and is formed in this step (Zhang et al, 2017, 2019). Splicing factors SRRM2, CWC22, and EJC are linked to sequences upstream of exon 5', where SRRM2 stabilizes the binding of exon 5' to the I loop of U5 snRNA, whereas CWC22 stabilizes SRRM2 and EJC (Galej et al, 2016; Zhang et al, 2017, 2019). The extended α-helix of

PRKRIP1 bridges the active splicing site with the core of U2 snRNP, with its N- and C-terminal portions interacting with the BPS/U2 duplex, suggesting they might be playing an important role in the C*/P complex by stabilizing the conformation of the active site (Zhang et al, 2019) (Fig 5). The ATPase/helicase DHX8 is prominently anchored in the binding domain of PRPF8 (Galej et al, 2016; Bertram et al, 2017b; Zhang et al, 2019). In the peripheral region of the P complex, SNRNP200 interacts with the COPS5 (Jab1)/MPN domain of PRPF8, and SNRNP200/COPS5 complex is connected to the spliceosomal nucleus through interactions with the stage II splicing factor SLU7 (Hegele et al, 2012; Zhang et al, 2019).

The transition from the P complex to the ILS is driven by DHX8, where the bound exon is released, causing an efflux of protein components and generating voids for subsequent spliceosome reorganization (Schwer, 2008; Zhang et al, 2019) (Fig 5). During the transition from P complex to ILS, dissociation of nine proteins was identified, with four as components of the EJC (eIF4AIII, MAGOH, MLN51, and Y14), the exon stabilizer protein SRRM2, the splicing factors CWC22 and SLU7, PRKRIP1, and ATPase/helicase DHX8 (Zhang et al, 2019) (Fig 5). Notably, CWF19L2 was recruited into the ILS1 complex and forms extensive interactions with PRPF8 and the intron lariat, which contribute to the stabilization of these molecules in ILS1 (Zhang et al, 2019). In addition, its structural characteristics support the idea that it assists in the debranching of the intron lariat of the RNA, in the BPS/U2 translocation, and in the disassembly of the spliceosome (Casalino et al, 2018) (Fig 5). Although its function is still unclear, it was observed that the inositol

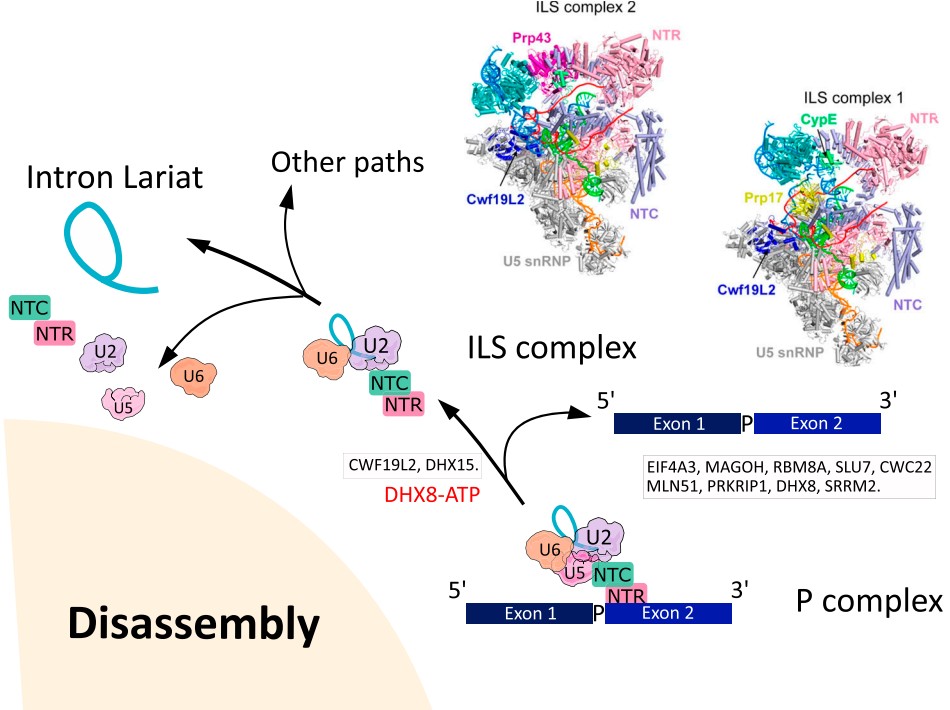

**Figure 5. Transition from the P complex to the ILS is driven by DHX8; the bound exon is released, causing an efflux of protein components and generating voids for subsequent spliceosome reorganization.** Notably, CWF19L2 was recruited into the ILS1 complex and contributed to the stabilization of ILS1; its structural characteristics support the idea that it assists in the debranching of the intron lariat of the RNA, in the BPS/U2 translocation, and in the disassembly of the spliceosome. ATPase/helicase DHX15 is loaded into the human ILS2 complex and mediates the disassembly of spliceosomes, which are recycled for the next round of splicing.

RODRIGUES, K. S. et al., (2022).

hexakisphosphate molecule (IP6) remains bound to PRPF8 during the P-to-ILS transition (Zhang et al, 2019). The ATPase/helicase DHX15 (Prp43) has not yet been loaded into the human ILS1 complex, which is the only difference between ILS1 and ILS2 (Liu et al, 2017; Wilkinson et al, 2017; Zhang et al, 2019). Thus, binding by DHX15 results in changes in the position of the surrounding components. DHX15 that is also linked to the intermediate portion of the NTC XAB2 (Syf1) component, the core of U2 snRNP, and the duplex BPS/U2 together with the splicing factor CDC40 in the ILS2 complex undergoes translocations in the ILS2 complex (Wan et al, 2017; Zhang et al, 2019) (Fig 5). The constant changing position of the U2 snRNP nucleus is a hallmark of spliceosome remodeling during each splicing cycle.

### IARA: An online atlas of curated and updated spliceosome-related molecules

All the data related to spliceosome machinery were curated and made available as an online atlas resource named IARA. The IARA atlas can be accessed by recent versions of most web browsers and mobile devices (Fig 6). The website has five main pages, namely, Home, About, Atlas, Collaborate, and Contact. The "Home" page provides a brief description of the site, its purpose, and information about the contents within the atlas. The "About" page contains a summary of the contributors of the atlas. The "Atlas" page presents the curated data, composed of the following fields in the main

table: Symbol, Name, Details, and Links. Also, the "Collaborate" page is available, allowing researchers to contribute, by sending updated information, novel molecules, novel annotations, or corrections to be included into the atlas. This page sends the data through a contact form, which informs the team responsible for verifying and validating the atlas information. And finally, the "Contact" page is also made up of a contact form with the team of collaborators.

The main functionality is available on page "Atlas." The end-user can interact with the atlas and search by gene Symbol or gene Name. Each curated gene has a button called Details, which opens up a new window for the visualization of additional information about the gene of the respective line, such as "Gene description on splicing," presenting a description of the gene in the splicing condition. Also, an image of a biological database of known and predicted protein–protein interactions obtained through an API (Application Programming Interface) is available, from STRING database, with the neighborhoods surrounding the protein from a related gene. The API works on HyperText Transfer Protocol requests based on the Uniform Resource Locator pattern (Table 1) and the values of the parameters (Table 2).

Other content provided by the "Details" button is the gene expression values, exon expression, and isoform expression by tissue queried on API from GTEx Portal (https://gtexportal.org/). The main functions used to get the information from the API (Table 3) are configured using specific parameters (Table 4).

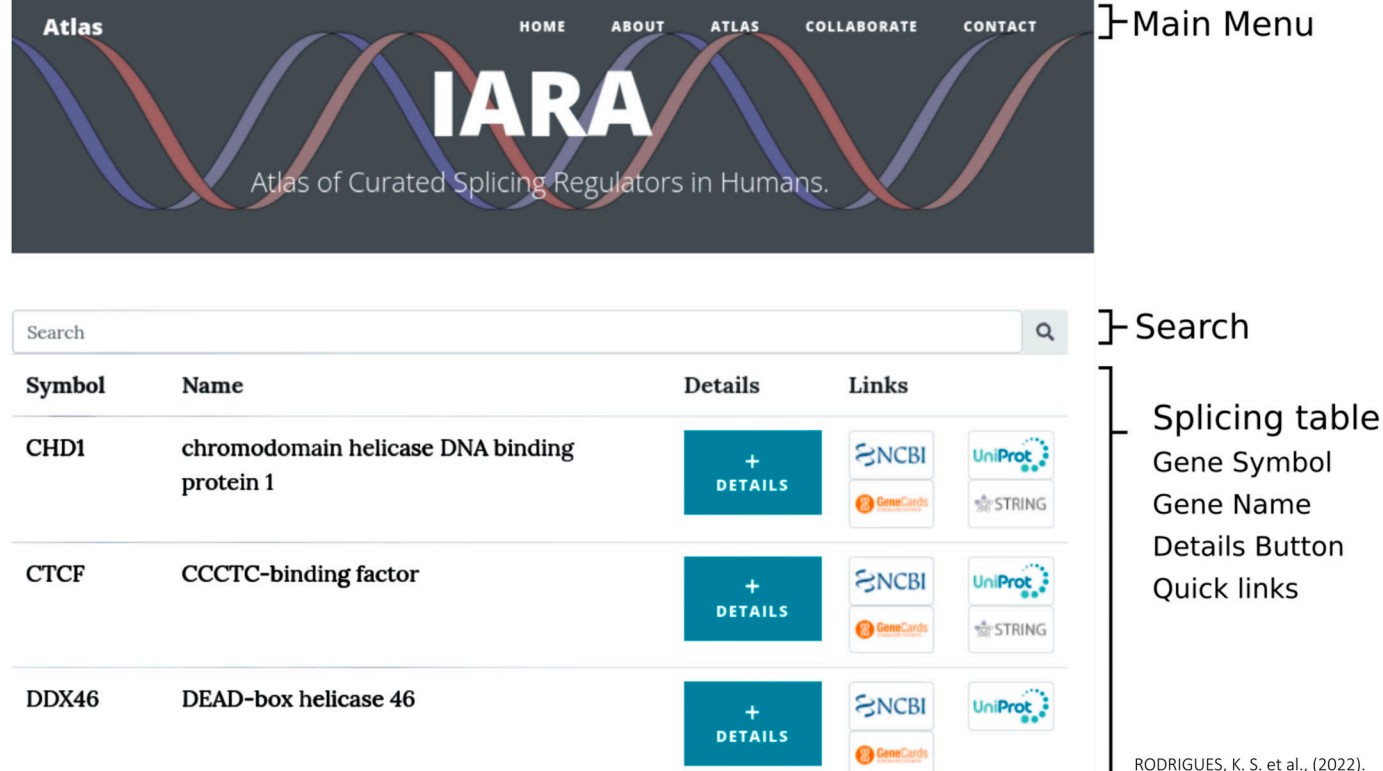

**Figure 6.  Website of the detailed and curated atlas of spliceosome-related molecules, IARA.**
The website is composed of three sections. The first section has the menu items to access the website functionalities; the second section has the search field to locate specific words or molecules within the entire website; the third section has the table that displays gene Symbol, gene Name, Details button, and quick Links for external databases.

**Table 1. URL path with parameters and description to the API function. URL patterns are prefixed by "https://string-db.org"**

| URL pattern | Parameters used | Description |
|---|---|---|
| /api/[output-format]/network | [output-format] [identifiers] [optional_parameters] | Retrieves an image of a STRING network of a neighborhood surrounding one or more proteins |

**Table 2. Parameters and respective values used to gather information from String-db.**

| Parameter | Description | Value |
|---|---|---|
| [output-format] | Output format: image (PNG image with alpha-channel), highres_image (high-resolution PNG image with alpha-channel), or SVG (Scalable Vector Graphics) | SVG |
| [identifiers] | Identifiers to gene or protein | Gene symbol |
| [optional_parameters] | None: loads default parameters (required_score = [depends on the network]; network_flavor = confidence; network type = functional; hide_node_labels = 0; hide_disconnected_nodes = 0; how_query_node_labels = 0; block_structure_pics_in_bubbles = 0) | None |

**Table 3. URL path with parameters and description to the API function. All URL patterns are prefixed by "https://gtexportal.org/rest/v1/"**

| URL pattern | Parameters used | Description |
|---|---|---|
| /reference/transcript | [datasetId] [gencodeId] | This service returns information about transcripts of the given versioned GENCODE ID |
| /reference/exon | [datasetId] [gencodeId] | This service returns exons from all known transcripts of the given gene |
| /expression/geneExpression | [datasetId] [gencodeId] [tissueSiteDetailId] [format] | This service returns normalized gene expression in tissues at the sample level |
| /expression/medianExonExpression | [datasetId] [hcluster] [gencodeId] [tissueSiteDetailId] [format] | This service returns median exon read counts by tissues of a given gene from all known transcripts |
| /expression/medianJunctionExpression | [datasetId] [hcluster] [gencodeId] [tissueSiteDetailId] [format] | This service returns median junction read counts by tissues of a given gene from all known transcripts |
| /expression/medianTranscriptExpression | [datasetId] [hcluster] [gencodeId] [tissueSiteDetailId] [format] | This service returns median normalized expression in tissues of all known transcripts of a given gene |

**Table 4. Parameters and respective values used to gather information from GTEx Portal.**

| Parameter | Description | Value |
|---|---|---|
| [datasetId] | Unique identifier of a dataset. Usually includes a data source and data release | gtex_v8 |
| [gencodeId] | Versioned GENCODE ID of a gene | Ensembl code (from the selected row of the table) |
| [tissueSiteDetailId] | Tissue ID of the tissue of interest | None: all tissues |
| [hcluster] | Specifies whether or not to perform hierarchical clustering | true |
| [format] | Format of result | JSON |

In addition, at the Details window, it is possible to visualize the curated references and quick links to the gene information at NCBI, UniProt, GeneCards, and String-db.

# Discussion

Splicing is a crucial post-transcriptional step in RNA processing for transcriptome and proteome diversity. In our revision, we found 74 genes (based on cryo-EM detection) directly involved with the spliceosome machinery that were annotated and classified according to their role in primary transcript splicing. In addition, performing a literature review, we generated the annotation and manual curation of 85 splicing factors that made up the splicing atlas we proposed here. The molecules present in the atlas were classified according to the main complexes of the spliceosomes, E, A, pre-M, B, B[act], B*, C, C*, P, and ILS complexes, and their association with the five main snRNA molecules, the U1, U2, U4, U5, and U6. The

NTC (19 complexes) and NTR (19-related) complexes were also considered.

The created atlas is also composed of a high number of different protein factors, such as transcription factors, RNA-binding proteins (RBPs), and non-coding RNAs (ncRNAs). The RBPs are fundamental to the splicing process, as they recognize regulatory elements within the pre-mRNA transcript and are the main components of the spliceosome. Kinase and phosphatase splicing factors activated through cell signaling mechanisms can, in turn, modify the activity of RBPs, contributing to their regulation (Shi & Manley, 2007; Naro & Sette, 2013). The ncRNAs are another important class of molecules within this process, which can regulate, directly or indirectly, the AS, by interacting with pre-mRNA or regulating the activity of splicing factors (Pisignano & Ladomery, 2021). Furthermore, growing evidence has shown that long non-coding RNAs act to regulate AS through conformational regulation of chromatin, by RNA-DNA hybridization, RNA-RNA interactions, and modulating the activity of splicing factors by regulating the localization and the phosphorylation status of these molecules (Hutchinson et al, 2007; Tsai et al, 2010; Hage et al, 2014; Conn et al, 2017). Circular RNAs (circRNAs) are also a class of ncRNAs, generated by a non-canonical type of splicing, called back-splicing, in which the 5′ terminus of an exon upstream of the pre-mRNA is linked non-collinearly with the 3′ terminus from a downstream exon to generate circRNAs, which may affect the outcome of linear canonical splicing (Guerra et al, 2020; Shao et al, 2021). In addition to the competition mechanism for biogenesis with linear splicing, circRNA may also favor the AS preference of its host gene (Conn et al, 2017). MicroRNAs are small ncRNAs that act through imperfect base pairing with target mRNAs; thus, they can also play a role in indirect regulation of splicing by regulating the post-transcriptional expression of splicing factors, as reported in the case of miR-133, which alters the splicing of several mRNAs involved in muscle maturation (Boutz et al, 2007). It was also recently shown that mitochondrial RNA can be spliced through a spliceosome-mediated mechanism, but additional investigations are required to better understand whether mitochondrial DNA-related molecules can also participate in this process (Herai et al, 2017). All these classes of molecules make the splicing regulation mechanism even more complex and highlight the importance of further studies on splicing mechanisms that associate with specific cellular phenotypes and their role in human pathologies.

### Malfunction of splicing factors influences disease occurrence

Splicing factors play a critical role in several phenotypes of the human body, and malfunctions in these factors, including mutations or by alteration of function, without changes in overall expression levels, can result in the occurrence or progression of various diseases (Du et al, 2021).

Several mutations in splicing factors are linked to cancer, such as *SF3B1* (splicing factor 3b subunit 1), a gene that encodes the subunit 1 of splicing factor 3b, which is the most frequently mutated RNA splicing factor in cancer (Lieu et al, 2022). Guo et al (2022) verified that a mutation in *SF3B1* generated aberrant splicing in the gene *DLG1* (disks large MAGUK scaffold protein 1) and thus, by activating the PI3K/Akt pathway, resulted in the progression of tumor invasion into prolactinoma, an adenoma in the pituitary gland (Guo et al,

2022). In another study, it was demonstrated that a mutation in *SF3B1* generated a missplicing in *MAP3K7* (mitogen-activated protein kinase kinase kinase 7), which resulted in a reduction in its expression level, generating severe anemia in patients with myelodysplastic syndrome (Lieu et al, 2022). It was also demonstrated that a mutant *SF3B1* recognizes an aberrant, deep intronic branch point within *BRD9* and thereby induces the inclusion of a poison exon that is derived from an endogenous retroviral element (Inoue et al, 2019). This process induces a subsequent degradation of *BRD9* mRNA, causing the loss of non-canonical BAF at *CTCF*-associated loci, promoting melanomagenesis (Inoue et al, 2019). In Shuai et al (2019), it was presented a highly recurrent A > C mutation at the third base of U1 snRNA; this changes the base pairing between U1 snRNA and the 5′ splice site, causing missplicing of pre-mRNAs of cancer driver genes (Shuai et al, 2019).

Also, dysregulation in splicing factors, such as the *U2AF1* (U2 small nuclear RNA auxiliary factor 1) gene, is related to hematological malignancies (Zhang et al, 2021a). It was also demonstrated that *U2AF1* mutations alter 3′ splice site recognition in hematological malignancies (Ilagan et al, 2015). The same splicing factor was reported by another study, which identified that low levels of *U2AF1* mRNA can be used as a prognosis for risk stratification in children with T-lineage acute lymphoblastic leukemia (Zhang et al, 2021a). It is worth mentioning another study, which also demonstrated that *U2AF1* causes intron retention in *CPNE1* (Copine 1), contributing to cellular senescence (Yao et al, 2020).

Neurodegenerative and neurodevelopmental diseases are also related to the dysregulation of splicing factors. It was shown that levels of U1 snRNA and vU1 snRNA (a variant of U1 snRNA) have a critical role in neuronal development and that changes in these levels contribute to the pathophysiology of motor neurons in patients with spinal muscular atrophy (Vazquez-Arango et al, 2016). In addition, another study also found a proteinopathy in U1 snRNP and several abnormal RNA splicing in patients with Alzheimer's disease (Bai et al, 2013).

Alterations in spliceosome-related genes were also found in Rett syndrome, a neurodevelopmental disorder caused by mutations in the methyl-CpG–binding protein 2 (*MECP2*) (Osenberg et al, 2018). Previous studies in mice showed that the loss of MECP2 function leads to dysregulation in AS during neuronal stimulation (Osenberg et al, 2018). Another study revealed that MECP2 forms a protein complex with Rbfox/LASR, and Rett mouse models with dysfunction in MECP2 interfere in the interaction of the MECP2/RBFOX/LASR complex, reducing RBFOX protein binding to specific pre-mRNA targets, and thus generating aberrant splicing events in Neurexins (NRXNS) and Neuroligin 1 (NLGN1) (Jiang et al, 2021). These genes encode transmembrane adhesion proteins that play a critical role in the plasticity of synapses (Jiang et al, 2021). In human cells, it has already been verified that the decrease in methylation caused by the reduction in MECP2 occupation to DNA reduced AS events and increased intron retention events (Wong et al, 2017).

## Conclusion

The splicing mechanism is a very important post-transcriptional molecular regulation that defines how pre-mRNA transcripts are processed to generate mature mRNAs. The entire mechanism is

regulated by the spliceosome machinery, and although its biogenesis is widely discussed throughout several independent works, it is still inconclusive how the entire mechanism works, including the complete list of molecules that makes up part of its regulation.

Previous studies, including literature reviews and online resources, made available a compilation of the involved molecules in splicing regulation, such as the online resources SpliceAid-F (Giulietti et al, 2013), Spliceosome Database (Cvitkovic & Jurica, 2013), and Reactome pathway Knowledgebase (Jassal et al, 2020). However, the available datasets are outdated, and in most cases, they do not apply a curation method to ensure that each listed molecule has a reliable role in splicing regulation and in which specific spliceosome-related cycle it is part of. Here, we used distinct strategies to collect reliable data from specialized literature to create an atlas, named as IARA, of all spliceosome-related molecules at cycle-level organization. In addition, all molecules were curated using an in silico literature-based cross-validation approach to ensure that all listed molecules have a specific role in each cycle of spliceosome machinery. IARA was developed as a website with a dynamic and automatic computational mechanism to keep it updated and to easily incorporate novel molecules once they are validated by our curation method.

## Materials and Methods

### Literature review

The literature review was performed in two steps. In the first one, we performed a search on PubMed for articles that applied cryo-EM methods to the structural study of the spliceosome in humans. The used keywords are "Cryo-EM," "Spliceosome," "human," and corresponding synonyms. In the second step, we performed a search on PubMed for articles using the keywords "human," "splicing," "gene," and corresponding synonyms. Next, after article data extraction, we integrated the extracted information to identify all key steps along with the splicing mechanism having distinct parts of the complexes forming the spliceosome machinery.

### Data curation

For the data curation step, we performed a critical review of the articles collected during the literature review step. These articles were used to perform a cross-validation approach for gene curation. For all genes identified as a candidate to be part of the curated atlas, the role of each one in splicing should be supported by at least two works of scientific literature. Moreover, all curated genes were also classified according to the stage they are involved in spliceosome machinery. For this step, we used scientific literature and the online tool REACTOME (available at URL: https://reactome.org/), a database with manual curation and peer-reviewed data validation.

### Online resource for splicing regulators

The atlas of splicing-related molecules was developed as a static website, and it is hosted on a public repository on GitHub (https://github.com/PUCPR-BioInformatics/atlas). The website is available through a static site hosting service, GitHub Pages (https://docs.github.com/en/pages), that runs a build process to publish its content. The build process uses Jekyll (https://jekyllrb.com/docs/) as a site generator that takes HyperText Markup Language, ECMAScript (also known as JavaScript), and Cascading Style Sheets to generate a complete website. Jekyll is composed of layouts and template engines that parse and compile structured information as comma-separated values and an API to consume web services to gather information from other services and render the webpage.

The site was hosted with the technology of GitHub pages, which is a source code repository platform and allows the creation of static sites, where the source code is publicly available, thus facilitating community-based collaboration, being possible to submit corrections, include novel add-ons, and include new curated genes per review check on a public GitHub repository.

## Data Availability

The atlas of splicing-related molecules was developed as a static website, and it is hosted on a public repository on GitHub (https://github.com/PUCPR-BioInformatics/atlas). The website is available through a static site hosting service, GitHub Pages (https://docs.github.com/en/pages), that runs a build process to publish its content.

## Supplementary Information

## Acknowledgements

The authors acknowledge the Pontifícia Universidade Católica do Paraná (PUCPR) for the structure to accomplish this review, the Fundação Araucária (FA, Paraná, Brazil), the Coordenação de Aperfeiçoamento de Pessoal de Nível Superior (CAPES, Brazil)—Finance Code 001—and Conselho Nacional de Desenvolvimento Científico e Tecnológico (CNPq, Brazil) for financing this study.

### Author Contributions

KS Rodrigues: conceptualization, data curation, formal analysis, methodology, and writing—original draft, review, and editing.
LP Petroski: conceptualization, software, formal analysis, methodology, and writing—review and editing.
PH Utumi: writing—original draft.
A Ferrasa: writing—original draft.
RH Herai: conceptualization, data curation, formal analysis, supervision, funding acquisition, validation, investigation, visualization, methodology, project administration, and writing—original draft, review, and editing.

## Conflict of Interest Statement

The authors declare that they have no conflict of interest.

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
