## [Reviewer comments · Life Science Alliance]

Life Science Alliance

IARA: A complete and curated atlas of the biogenesis of Spliceosome machinery during RNA splicing.

Kelren Rodrigues, Luiz Petroski, Paulo Utumi, Adriano Ferrasa, and Roberto Herai

DOI: <https://doi.org/10.26508/lsa.202201593>

Corresponding author(s): Roberto Herai, Pontificia Universidade Catolica do Parana (PUCPR), Graduate Program in Health Sciences (PPGCS).

Review Timeline:

Submission Date:	2022-07-06
Editorial Decision:	2022-07-28
Revision Received:	2022-10-07
Editorial Decision:	2022-10-21
Revision Received:	2022-12-08
Accepted:	2022-12-08

Scientific Editor: Novella Guidi

Transaction Report:

July 28, 2022

Re: Life Science Alliance manuscript #LSA-2022-01593-T

Roberto Hirochi Herai
Pontificia Universidade Catolica do Parana (PUCPR), Graduate Program in Health Sciences (PPGCS).

Dear Dr. Herai,

Thank you for submitting your manuscript entitled "IARA: A curated genetic atlas of human spliceosome machinery" to Life Science Alliance. The manuscript was assessed by expert reviewers, whose comments are appended to this letter. We invite you to submit a revised manuscript addressing the Reviewer comments.

Thank you for this interesting contribution to Life Science Alliance. We are looking forward to receiving your revised manuscript.

Sincerely,

B. MANUSCRIPT ORGANIZATION AND FORMATTING:

Reviewer #1 (Comments to the Authors (Required)):

The manuscript titled "IARA: A curated genetic atlas of human spliceosome machinery" presents a bioinformatics tool that maps the steps involved in the human splicing mechanism, and it allows a detailed overview of the molecules involved throughout the distinct steps of splicing. This tool can be very useful for the scientific community. It is a curated atlas of spliceosome-related molecules is crucial to provide all researchers a reliable resource to better investigated this important mechanism. There are only minors review to be done in this manuscript. The information about the subject (human spliceosome machinery) is well written and the presentation of the new tool well explained. The conclusion is concise and understandable.

Page 5 - paragraph 3 "where the source code of the ... is publicly available" something is missing here.

Page 6 - paragraph 1 "A forms the pre-B complex. At this time, U1 snRNA is still paired to the 5'ss, and the U2/U6 helix forms (Agafonov et al.)." correct the reference.

There is no citation of figure 2 in the text.

Page 19 - What does Luiz mean?

Reviewer #2 (Comments to the Authors (Required)):

In this manuscript, Rodrigues et al. compiled an up-to-date list of the human spliceosomal genes involved in different steps of the splicing process through a literature review followed by a manual curation. Primarily based on information in the published Cryo-EM structures of the human spliceosome, the authors presented in an organized manner the highly dynamic protein and RNA composition of the human spliceosome during its assembly, activation, splicing catalysis, and disassembly. The authors also created an online resource summarizing the known functions, expression patterns, and protein-protein interactions of the spliceosomal genes with the external links and references included. And finally, the authors tried to signify their work by reviewing some examples of defective splicing factors in splicing misregulation in neurological diseases.

Overall, the authors made a good effort to summarize the protein/RNA composition of the human spliceosome in each step of splicing. However, there have been numerous reviews of the multiple steps of the splicing process. And the proteins and RNAs in the human spliceosomal complexes have already been known in published studies of the Cryo-EM structures, the mere effort of compiling them together into this study is not inspiring. Therefore, the significance of this study is very limited.

Specific points:

1. Although the authors aimed to provide the most up-to-date list of the genes in the human spliceosome, they did not use the most updated human gene names in the main text. The authors did provide the updated human gene names in Figure 1C and on their website. However, it is important that the authors are consistent and use the most updated human gene names in the main text as well.
2. One major issue of this study is that the authors did not include in the "atlas" the spliceosomal genes of Complexes E and A, which are very important in the early stages of spliceosome assembly. At least, the authors need to provide the protein/RNA compositions of the U1 and U2 snRNPs (see refs. Kondo et al., eLife 2015 and Zhang et al., Nature 2020). Also, in Figure 1, in addition to SF1, HTATSF1 (previously known as TAT-SF1) and DDX46 (Prp5) are also involved in U2-Branchsite recognition.
3. The authors used the external tool REACTOME in their website to summarize the protein-protein interactions of spliceosomal proteins. However, this tool missed some of the important interactions. For example, it is known that SF3B1 has multiple direct interactions with DDX46 and SF3B3, which are important for early stages of spliceosome assembly. However, these interactions are not included in the REACTOME. The authors may want to generate an accurate, complete, and up-to-date database of protein-protein interactions among the spliceosomal proteins based on the Cryo-EM structures.
4. The authors wrote at the end of this manuscript a few examples of defective splicing factors (TDP-43, FUS, SMN1, DMD, MECP2, RBFOX, and MAPT) in splicing misregulation in neurological disorders to demonstrate the significance of RNA missplicing to disease. However, none of these are among the spliceosomal genes shown in Figure 1. Therefore, this section of text seems irrelevant. I recommend that the authors include the more relevant spliceosomal genes (for example SF3B1, U2AF1, and U1 snRNA) that are frequently mutated in a wide variety of hematological malignancies and solid tumors. The mechanisms of these spliceosomal gene mutations in RNA missplicing in cancer have been well known. It would be more convincing that the authors include these mutated spliceosomal genes in this manuscript to demonstrate that spliceosomal gene mutations have

significant implications in disease.

Response to Reviewers

Reviewer #1

The manuscript titled "IARA: A curated genetic atlas of human spliceosome machinery" presents a bioinformatics tool that maps the steps involved in the human splicing mechanism, and it allows a detailed overview of the molecules involved throughout the distinct steps of splicing. This tool can be very useful for the scientific community. It is a curated atlas of spliceosome-related molecules is crucial to provide all researchers a reliable resource to better investigated this important mechanism. There are only minors review to be done in this manuscript. The information about the subject (human spliceosome machinery) is well written and the presentation of the new tool well explained. The conclusion is concise and understandable.

We thank the reviewer for the time spent revising our manuscript. As he pointed out, we also believe that a curated atlas of human spliceosome machinery can represent an important source of information for researchers involved in the field of transcription processing. In the following paragraphs, we addressed all the referee's concerns.

Page 5 - paragraph 3 "where the source code of the ... is publicly available" something is missing here.

As requested, the sentence has been corrected as follow: "where the source code is publicly available".

Page 6 - paragraph 1 "A forms the pre-B complex. At this time, U1 snRNA is still paired to the 5'ss, and the U2/U6 helix forms (Agafonov et al.)." correct the reference.

As requested, the reference has been corrected as follow: "(Agafonov et al., 2016)."

There is no citation of figure 2 in the text.

As requested, Figure 2 is now correctly cited within manuscript.

Page 19 - What does Luiz mean?

The word Luiz was removed.

Reviewer #2

In this manuscript, Rodrigues et al. compiled an up-to-date list of the human spliceosomal genes involved in different steps of the splicing process through a literature review followed by a manual curation. Primarily based on information in the published Cryo-EM structures of the human spliceosome, the authors presented in an organized manner the highly dynamic protein and RNA composition of the human spliceosome during its assembly, activation, splicing catalysis, and disassembly. The authors also created an online resource summarizing the known functions, expression patterns, and protein-protein interactions of the spliceosomal genes with the external links and references included. And finally, the authors tried to signify their work by reviewing some examples of defective splicing factors in splicing misregulation in neurological diseases.

Overall, the authors made a good effort to summarize the protein/RNA composition of the human spliceosome in each step of splicing. However, there have been numerous reviews of the multiple steps of the splicing process. And the proteins and RNAs in the human spliceosomal complexes have already been known in published studies of the Cryo-EM structures, the mere effort of compiling them together into this study is not inspiring. Therefore, the significance of this study is very limited.

We thank the reviewer for the time and effort spent with the manuscript's revision and valuable comments/suggestions. We agree with the reviewer that proteins and RNAs in the human spliceosomal complexes have already been known in

published studies of the Cryo-EM structures, however all the available studies provided a fragmented information of spliceosome machinery. Furthermore, after reviewing the current literature, we also found that is very difficult to correlate the molecules from different studies to have a global snapshot of all molecules involved in each step of splicing processing mechanism.

Thus, a web-resource with curated information and all stages of RNA splicing mechanism represents an important source of information for the researchers in the field of genetics and molecular biology.

Although the authors aimed to provide the most up-to-date list of the genes in the human spliceosome, they did not use the most updated human gene names in the main text. The authors did provide the updated human gene names in Figure 1C and on their website. However, it is important that the authors are consistent and use the most updated human gene names in the main text as well.

We thank the reviewer for the important observation. To address this concern, all gene names were revised and corrected within manuscript. Additionally, we also followed a standard nomenclature, with gene symbols followed by their corresponding informal names in parentheses on their first appearance in the text. All modifications are highlighted in blue within the manuscript.

One major issue of this study is that the authors did not include in the "atlas" the spliceosomal genes of Complexes E and A, which are very important in the early stages of spliceosome assembly. At least, the authors need to provide the protein/RNA compositions of the U1 and U2 snRNPs (see refs. Kondo et al., eLife 2015 and Zhang et al., Nature 2020). Also, in Figure 1, in addition to SF1, HTATSF1 (previously known as TAT-SF1) and DDX46 (Prp5) are also involved in U2-Branchsite recognition.

We thank the reviewer by pointing out this very important observation regarding the missing information in our manuscript. To address this concern, we included the two mentioned complexes (Complexes E and A), and the Pre-B complex which is also part of the initial steps of RNA splicing. As suggested, we also provided the protein/RNA compositions of the U1 and U2 snRNPs (refs. Kondo et al., eLife 2015 and Zhang et al., Nature 2020). Finally, we also updated the Figure 1, in addition to SF1, HTATSF1 and DDX46 (Prp5) were also included in the U2-Branchsite recognition. All these modifications were included within the manuscript, and are listed in the following paragraphs:

“E complex

During spliceosome assembly, the U1 snRNP complex interacts with a short sequence (nucleotides 3 to 10) at the 5' intron splice site and stabilizes base pairing with the 5' end of the U1 snRNA (Kondo et al. 2015; Will et al. 1996). In this step, U2AF and splicing factor 1 (SF1) bind to the 3' splicing site and branch point site, forming the spliceosome E complex (Kondo et al. 2015; Will et al. 1996). The composition and spatial organization of the human U1 snRNP functional nucleus was determined through electron density maps, with crystal structures of resolution of 5.5 Å, 4.4 Å, 3.3 Å, and 2.5 Å (Weber et al. 2010; Kondo et al. 2015; Pomeranz Krummel et al. 2009). These works demonstrated the minimal structure of U1 snRNP with the 5' splicing site RNA (Pomeranz Krummel et al. 2009); the U1 A70KF-RNA crystal structure of the SNRNP70 (U1-70k, residue 60-216) linked to stem-loop I of U1 snRNA (Kondo et al. 2015); and the organizational structure of the seven proteins of the Sm complex in snRNAs (Weber et al. 2010).

During the formation of the E complex, the U1 snRNA molecule forms four stem-loops (SL1, SL2, SL3, and SL4) and the H helix, a region formed by the pairing between nucleotides 12-16 and 118-122 (Weber et al. 2010). The snRNP U1 has a minimum structure of seven Sm proteins (SNRPB (Sm-B/Sm-B'), SNRPD1 (Sm-D1), SNRPD2 (Sm-D2), SNRPD3 (Sm-D3), SNRPE (Sm-E), SNRPF (Sm-F), and SNRPG (Sm-G)), three specific U1 proteins (SNRNP70 (U1-70K), SNRPA (U1-A), and SNRPC (U1-C)), and the U1 snRNA (Pomeranz Krummel et al. 2009). The seven Sm proteins are present in all spliceosome U snRNPs, except for U6 (Weber et al. 2010; Achsel et al. 1999). In U1 snRNP, the Sm proteins form in a heptameric ring, around the nucleotides of the Sm site (a conserved region rich in uracil), placed between the SL3 and SL4 in the U1 snRNA, forming the central domain of the U1 snRNP complex (Weber et al. 2010). The spatial structure of the RNA helices is organized into a SL4 located 3' of the Sm site and four 5' helices, with SL1 and SL2 coaxially stacked, as well as SL3 and H helix (Weber et al. 2010). The SNRNP70 and SNRPA proteins interact with SL1 and SL2 through their RNA-binding domain (Oubridge et al. 1994; Pomeranz Krummel et al. 2009). SNRPC stabilizes base pairing between the 5' splice site of the pre-mRNA and the 5' end of the U1 snRNA, forming hydrogen bonds with the 2'OH groups and with phosphate atoms of both nearby RNA strands at the splicing junction (Kondo et al. 2015; Bertram et al. 2017a).

The 5' end of the U1 snRNA and the general structure of the U1 snRNP are stabilized by the N-terminal helices of SNRPD2 and SNRPB in a particular orientation relative to the Sm ring (Pomeranz Krummel et al. 2009). SNRPD2

binds with the H helix and has conserved regions that enable N-terminal interaction with RNA and N-terminal SNRPB also interacts with RNA at base SL2 (Pomeranz Krummel et al. 2009; Kondo et al. 2015) . The N-terminal region of the SNRNP70 protein may be involved in restricting the movement of SL1 in relation to the central domain (Pomeranz Krummel et al. 2009). The RBD domain of SNRNP70 surrounds the set of Sm proteins through a long α -helix and its N-terminal portion helps the binding of the SNRPC protein to the central Sm domain by interacting with SNRPD3 (Kondo et al. 2015). SNRPC is crucial for E complex formation as it binds to the minor groove of the RNA duplex by pairing with the GU invariant portion of the 5' junction site and its zinc finger domain stabilizes the U1 RNA duplex snRNA and the 5' junction site (Kondo et al. 2015; Pomeranz Krummel et al. 2009). The SNRPA protein binds to SL2 through its N-terminal RBD domain and positions it to interact with SNRPB and SNRPD1 in the Sm ring, making the RNA structure in the central domain even more stable (Kondo et al. 2015; Pomeranz Krummel et al. 2009).

A complex

After U1 complex formation in the 5' intron splice site, the A complex is assembled. At this point, the snRNAs U2 interact with the 5' junction site (ss) and the branching site (BS) of the intron (Wahl et al. 2009; Krämer et al. 1984; Black et al. .1985) (Figure 2). The molecular architecture of the human U2 complex was described using 3D cryoelectronic microscopy (3D cryo-EM) and protein crosslinking data. Zhang et al (2020), determined the structure of the main subunit of A complex, the human 17S U2 snRNP (4.1 Å), exhibiting a bipartite

organization of two domains, divided into a major U2 5' module and a minor U2 module. 3' (Zhang et al. 2020). The 5' 17S U2 domain has the SF3B complex (SF3B3, PHF5A, SF3B5 and SF3B1^{HEAT}) as its main component and is connected to the 3' domain, which has the U2 Sm RNP core linked by U2-A' and U2-B'' to the U2 snRNA (Zhang et al. 2020). The SF3A complex (SF3A3 (SF3a60), SF3A2 (SF3a66) and SF3A1 (SF3a120)) is part of the 3' domain and is involved in the bridge between the U2 Sm core and SF3B (Zhang et al. 2020). Subsequently, Cretu and colleagues (2021), also presented a 3D Cryo-EM structure (~3.1 Å central resolution) of the U2 5' module and described the SF3B complex, two zinc finger domains of SF3A2 and SF3A3 and part of the substrate of intron paired with the U2 snRNA (Cretu et al. 2021).

Through an ATP-dependent reaction, the displacement of SF1 occurs, which enables the recruitment of U2 snRNP (Gao et al. 2008; Taggart et al. 2017). According to the proposed structures, U2 snRNP is essential for the accurate recognition of BS within the intron by the loop region called branchpoint-interacting-stem-loop (BSL) in the U2 snRNA, forming the U2/intron duplex (van der Feltz and Hoskins 2019; Zhang et al. 2020). The transition from BSL to the U2/intron duplex is promoted by the action of PRP5/DDX46, HTATSF1, and SF3A2 molecules (Cretu et al. 2021; Perriman and Ares 2010). PRP5/DDX46 acts to stabilize the incorporation of U2 into the A complex and facilitate changes in the structure of U2 snRNP (Zhang et al. 2020). For this incorporation to occur, HTATSF1 is displaced from SF3B1 by the action of PRP5/DDX46, releasing BSL for the transition (Zhang et al. 2020; Cretu et al. 2021). It is also possible that in

the BS recognition process, PRP5/DDX46 is acting on U2 snRNA remodeling (Zhang et al. 2020). After the release of BSL, SF3A2 binds to the precursor helix aiding in its formation and stabilization. PRP5 covers SF3B1^{HEAT}, and assists in maintaining its conformation, which adopts an open state in 17S U2 snRNP, which becomes closed to stabilize the extended U2/intron helix near the 3'ss (A-to-B^{act} spliceosome) (Bertram et al. 2017a; Cretu et al. 2021). The SF3B5, PHF5A, and SF3B3 subunits are similarly organized (Bertram et al. 2017a; Cretu et al. 2021).

In the following steps, the spliceosome goes through 8 functional states that are divided into: precursor of the pre-catalytic spliceosome (pre-B), pre-catalytic spliceosome (B), activated spliceosome (B^{act}), catalytically activated spliceosome (B*) (Bertram et al. 2017a; Zhang et al. 2018; Agafonov et al. 2016), catalytic step I complex (C), step II catalytically activated complex (C*) (Zhan et al. 2018a; Zhang et al. 2017; Bertram et al. 2017b), post-catalytic spliceosome (P), and intron lariat spliceosome (ILS) (Zhang et al. 2019).

Pre-B complex

The initial and still unstable binding of tri-snRNP (U4/U6.U5) with U2 snRNP to complex A forms the pre-B complex. The U1 snRNA is still paired with the 5'ss and the U5 and U6 snRNAs have not yet recognized the pre-mRNA (Agafonov et al. 2016). After the association of the tri-snRNP, the transcript undergoes conformational changes that promote the formation of the U2/U6 duplex, the release of U1 mediated by the DDX23 helicase (Prp28), and the

rearrangement of the SF3A and SF3B complexes of the tri-snRNP, generating the B complex (Zhan et al. 2018b; Bertram et al. 2017a) (Figure 3). The cryo-MS structure of the human pre-B complex (5.7 Å) demonstrated that the tetrahedron conformation of the tri-snRNP is composed of four vertices: the U4 Sm, U5 Sm, U6 LSm ring, and the SNRNP200 helicase (Brr2) (Zhan et al. 2018b). Between the tetrahedron edges of the U4 Sm and U6 LSm rings is the U2 snRNP, while the U1 snRNA is between the Sm-4 and Sm-5 rings (Zhan et al. 2018b). The pre-B spliceosome complex is the only one that has the five snRNPs present, formed by U1 and U2 snRNP, tri-snRNP (PRPF3 (Prp3), PRPF4 (Prp4), PRPF6 (Prp6), PRPF31 (Prp31), SNU13 (Snu13), TXNL4A (Dim1), USP39 (Sad1)), U5 snRNP (PRPF8 (Prp8), EFTUD2 (Snu114), SNRNP200, SNRNP40 (U5-40k), U5 Sm, and U5 snRNA), U4 snRNP (U4 Sm and U4 snRNA), U4 snRNP (U6 Sm, and U6 snRNA), and DDX23 (Zhan et al. 2018b).

The previously formed U2 intron/snRNA duplex is still present in the pre-B complex, and sequences from the 3' end of the U6 snRNA to the 5' end of the U2 snRNA form helix II (Zhan et al. 2018b; Anokhina et al. 2013). Also, U2 snRNP binds to tri-snRNP through interactions between the SF3B complex and the U6 LSm ring (Zhan et al. 2018b). The central nucleus shared between pre-B and B belongs to the U5 snRNP complex, having an almost invariable conformation, supported mainly by the N-terminal domain of PRPF8, together with EFTUD2, SNRNP40, the U5 Sm ring and the U5 snRNA molecule (Hang et al. 2015). The N-terminal domain of PRPF8 interacts with the helicase DDX23 (linked to U1 snRNP), while its C-terminal domain Jab1 link the tri-snRNP nucleus to the

SNRNP200 protein (Mozaffari-Jovin et al. 2013; Nguyen et al. 2013). SNRNP200 also establishes this link through PRPF6 and USP39 (Hang et al. 2015). The PRPF6 protein, forms two short α -helices on the exposed surface of TXNL4A that interact directly with loop I of U5 snRNA (Zhan et al. 2018b). The transfer of base pairing between U1/5'ss snRNA to U6/5'ss snRNA starts the pre-B to B transition, which triggers further unwinding of the U4/U6 duplex by the SNRNP200 helicase (Zhan et al. 2018b). For this transition to occur, most tri-snRNP components undergo structural and organizational changes, including some PRPF8 domains (Zhan et al. 2018b). The dissociation of specific pre-B proteins (DDX23 and USP39) and the recruitment of new molecules to the B complex is also triggered (Zhan et al. 2018b).”

The authors used the external tool REACTOME in their website to summarize the protein-protein interactions of spliceosomal proteins. However, this tool missed some of the important interactions. For example, it is known that SF3B1 has multiple direct interactions with DDX46 and SF3B3, which are important for early stages of spliceosome assembly. However, these interactions are not included in the REACTOME. The authors may want to generate an accurate, complete, and up-to-date database of protein-protein interactions among the spliceosomal proteins based on the Cryo-EM structures.

We thank the reviewer for the important observation on protein-protein interaction limitations provided by REACTOME. However, our aim on using REACTOME was to only classify the proteins of the spliceosome complexes (For example: B complex, Batc, etc...). Thus, a protein-protein interaction network between spliceosome-related molecules is not part of our atlas since there are other

available tools that can provide that information (Ingenuity, PathwayCommons, etc.).

The authors wrote at the end of this manuscript a few examples of defective splicing factors (TDP-43, FUS, SMN1, DMD, MECP2, RBFOX, and MAPT) in splicing misregulation in neurological disorders to demonstrate the significance of RNA missplicing to disease. However, none of these are among the spliceosomal genes shown in Figure 1. Therefore, this section of text seems irrelevant. I recommend that the authors include the more relevant spliceosomal genes (for example SF3B1, U2AF1, and U1 snRNA) that are frequently mutated in a wide variety of hematological malignancies and solid tumors. The mechanisms of these spliceosomal gene mutations in RNA missplicing in cancer have been well known. It would be more convincing that the authors include these mutated spliceosomal genes in this manuscript to demonstrate that spliceosomal gene mutations have significant implications in disease.

We thank the reviewer for the suggestion. As we believe a broader description of distinct diseases correlated with spliceosome, we included a novel topic discussing several spliceosomal-related genes from Figure 1 that are involved in several diseases, including cancer, hematological diseases, and neurological disorders. The included topic within manuscript is also listed below:

“Malfunction of Splicing Factors influences diseases occurrence

Splicing factors play a critical role in several phenotypes of human body, and malfunctions in these factors can result in the occurrence or progression of various diseases (Du et al. 2021).

Several mutations in splicing factors are linked to cancer, among them, SF3B1 (Splicing Factor 3b Subunit 1), a gene that encodes the subunit 1 of splicing factor 3b, is the most frequently mutated RNA splicing factor in cancer

(Lieu et al. 2022). Guo et. al. (2022) verified that a mutation in SF3B1 generated aberrant splicing in the gene DLG1 (Discs Large MAGUK Scaffold Protein 1) and thus, activating the PI3K/Akt pathway, resulting in the progression of tumor invasion into prolactinoma, an adenoma in the pituitary gland (Guo et al. 2022). In another study, it was demonstrated that a mutation in SF3B1 generated a mis-splicing in MAP3K7 (Mitogen-Activated Protein Kinase Kinase Kinase 7), which resulted in a reduction in its level, generating severe anemia in patients with myelodysplastic syndrome (Lieu et al. 2022).

Also, dysregulation in splicing factors, such as the U2AF1 (U2 Small Nuclear RNA Auxiliary Factor 1) gene, is related to hematological diseases (Zhang et al. 2021a). The same splicing factor was reported by another study, that identified that low levels of U2AF1 mRNA can be used as a prognosis for risk stratification in children with T-lineage acute lymphoblastic leukemia (Zhang et al. 2021a). It is worth mentioning another study that also demonstrated that U2AF1 causes intron retention in CPNE1 (Copine 1), contributing to cellular senescence (Yao et al. 2020).

Neurodegenerative and neurodevelopmental diseases are also related to dysregulation of splicing factors. It was demonstrated that levels of U1 snRNA and vU1 snRNA (a variant of U1 snRNA) have a critical role in neuronal development, and that changes in these levels, contribute to the pathophysiology of motor neurons in patients with spinal muscular atrophy (Vazquez-Arango et al. 2016). Additionally, another study also found a

proteinopathy in U1 snRNP and several abnormal RNA splicing in patients with Alzheimer's disease (Bai et al. 2013).

Alterations in spliceosome-related genes were also found in Rett syndrome, a neurodevelopmental disorder caused by mutations in the methyl-CpG binding protein 2 (MECP2) (Osenberg et al. 2018). Previous studies in mice showed that loss of MECP2 function leads to dysregulation in alternative splicing during neuronal stimulation (Osenberg et al. 2018). Another study revealed that MECP2 forms a protein complex with Rbfox/LASR, and Rett mice models with dysfunction in MECP2 interfere in the interaction of MECP2/RBFOX/LASR complex, reducing RBFOX protein binding to specific pre-mRNA targets, and thus generating aberrant splicing events in Neurexins (NRXNS) and Neuroligin1 (NLGN1) (Jiang et al. 2021). These genes encode to transmembrane adhesion proteins that play a critical role in the plasticity of synapses (Jiang et al. 2021). In human cells, it has already been verified that the decrease in methylation caused by the reduction of MECP2 occupation to DNA reduced alternative splicing events, and increasing intron retention events (Wong et al. 2017).”

October 21, 2022

RE: Life Science Alliance Manuscript #LSA-2022-01593-TR

Dr. Roberto Hirochi Herai
Pontificia Universidade Catolica do Parana (PUCPR), Graduate Program in Health Sciences (PPGCS).
School of Medicine and Life Sciences
R. Imac. Conceição, 1155 - Prado Velho, Curitiba - PR, 80215-901, PPGCS
Curitiba, Paraná 80215901
Brazil

Dear Dr. Herai,

Thank you for submitting your revised manuscript entitled "IARA: A complete and curated atlas of the biogenesis of Spliceosome machinery during RNA splicing.". We would be happy to publish your paper in Life Science Alliance pending final revisions necessary to meet our formatting guidelines.

- please include the few important studies as requested by Reviewer 2
- please upload your manuscript text as an editable doc file
- please add the Author Contributions to the main manuscript text
- please consult our manuscript preparation guidelines <https://www.life-science-alliance.org/manuscript-prep> and make sure your manuscript sections are in the correct order
- please upload your Figure 6 as a single file
- please provide a separate Data availability section after Materials and Methods

A. FINAL FILES:

B. MANUSCRIPT ORGANIZATION AND FORMATTING:

Sincerely,

Reviewer #2 (Comments to the Authors (Required)):

In this revision, the authors addressed the reviewers' questions and improved the manuscript significantly. The revised manuscript compiled an accurate list of genes in the human spliceosome at different stages of the splicing process, and therefore is of value to the scientific community.

One minor issue still needs to be addressed. The authors revised a large section of the text that described the disease relevance of malfunction of splicing factors. However, a few important studies were not included. Some spliceosomal genes are frequently mutated in cancer and the mutated spliceosomal factors can contribute to disease by alteration-of-function, without changes of overall expression levels. For example: a recurrent A>C mutation at the third base of U1 snRNA changes the base-pairing between U1 snRNA and the 5' splice site, leading to missplicing of pre-mRNAs of cancer driver genes (see Shuai, S. et al. 2019. Nature, 574: 712-716). SF3B1 mutations lead to the inclusion of a poison exon that disrupts translation of BRD9, promoting melanomagenesis (see Inoue, D. et al. 2019. Nature, 574: 432-436). U2AF1 mutations alter 3' splice site recognition in hematological malignancies (see Ilagan J.O. et al. 2015. Genome Res, 25: 14-26). These studies can be included in this manuscript to signify the authors' efforts of compiling the list of spliceosomal genes.

Response to Reviewer

Reviewer #2 (Comments to the Authors (Required)):

In this revision, the authors addressed the reviewers' questions and improved the manuscript significantly. The revised manuscript compiled an accurate list of genes in the human spliceosome at different stages of the splicing process, and therefore is of value to the scientific community.

We thank the reviewer for the time spent revising our manuscript. Our study has several information that are significant and important to the scientific community that aims to study splicing in humans, but also in other organisms.

One minor issue still needs to be addressed. The authors revised a large section of the text that described the disease relevance of malfunction of splicing factors. However, a few important studies were not included. Some spliceosomal genes are frequently mutated in cancer and the mutated spliceosomal factors can contribute to disease by alteration-of-function, without changes of overall expression levels. For example: a recurrent A>C mutation at the third base of U1 snRNA changes the base-pairing between U1 snRNA and the 5' splice site, leading to missplicing of pre-mRNAs of cancer driver genes (see Shuai, S. et al. 2019. Nature, 574: 712-716). SF3B1 mutations lead to the inclusion of a poison exon that disrupts translation of BRD9, promoting melanomagenesis (see Inoue, D. et al. 2019. Nature, 574: 432-436). U2AF1 mutations alter 3' splice site recognition in hematological malignancies (see Ilagan J.O. et al. 2015. Genome Res, 25: 14-26). These studies can be included in this manuscript to signify the authors' efforts of compiling the list of spliceosomal genes.

We agree with the reviewer that the inclusion of novel references would be of great interest for the readers. Thus, as suggested, we included the following paragraphs within our manuscript.

Splicing factors play a critical role in several phenotypes of human body, and malfunctions in these factors, including mutations or by alteration-of-function, without changes of overall expression levels, can result in the occurrence or progression of various diseases (Du et al., 2021).

Several mutations in splicing factors are linked to cancer, such as *SF3B1* (Splicing Factor 3b Subunit 1), a gene that encodes the subunit 1 of splicing factor 3b, which is the most frequently mutated RNA splicing factor in cancer (Lieu et al., 2022a). Guo et al. (2022) verified that a mutation in *SF3B1* generated aberrant splicing in the gene *DLG1* (Discs Large MAGUK Scaffold Protein 1) and thus, by activating the PI3K/Akt pathway, resulted in the progression of tumor invasion into prolactinoma, an adenoma in the pituitary gland (Guo et al., 2022). In another study, it was demonstrated that a mutation in *SF3B1* generated a mis-splicing in *MAP3K7* (Mitogen-Activated Protein Kinase Kinase Kinase 7), which resulted in a reduction in its expression level, generating severe anemia in patients with myelodysplastic syndrome (Lieu et al., 2022b). In another study, it was demonstrated that a mutant *SF3B1* recognizes an aberrant, deep intronic branchpoint within *BRD9* and thereby induces the inclusion of a poison exon that is derived from an endogenous retroviral element (Inoue et al., 2019). This process induces a subsequent degradation of *BRD9* mRNA, causing the loss of non-canonical BAF at CTCF-associated loci, promoting melanomagenesis (Inoue et al., 2019). In another study, Shuai et al. (2019) present a highly recurrent A>C mutation at the third base of U1 snRNA, this changes the base-pairing between U1 snRNA and the 5' splice site, causing missplicing of pre-mRNAs of cancer driver genes (Shuai et al., 2019).

Also, dysregulation in splicing factors, such as the *U2AF1* (U2 Small Nuclear RNA Auxiliary Factor 1) gene, is related to hematological malignancies (P. Zhang et al., 2021a). It was also demonstrated that *U2AF1* mutations alter 3' splice site recognition in hematological malignancies (Ilagan et al., 2015). The same splicing factor was reported by another study, that identified that low levels of *U2AF1* mRNA can be used as a prognosis for risk stratification in children with T-lineage acute lymphoblastic leukemia (P. Zhang et al., 2021b). It is worth mentioning another study that also demonstrated that *U2AF1* causes intron retention in *CPNE1* (Copine 1), contributing to cellular senescence (Yao et al., 2020).

December 8, 2022

RE: Life Science Alliance Manuscript #LSA-2022-01593-TRR

Dr. Roberto Hirochi Herai
Pontificia Universidade Catolica do Parana (PUCPR), Graduate Program in Health Sciences (PPGCS).
School of Medicine and Life Sciences
R. Imac. Conceição, 1155 - Prado Velho, Curitiba - PR, 80215-901, PPGCS
Curitiba, Paraná 80215901
Brazil

Dear Dr. Herai,

Thank you for submitting your Research Article entitled "IARA: A complete and curated atlas of the biogenesis of Spliceosome machinery during RNA splicing.". It is a pleasure to let you know that your manuscript is now accepted for publication in Life Science Alliance. Congratulations on this interesting work.

DISTRIBUTION OF MATERIALS:

Again, congratulations on a very nice paper. I hope you found the review process to be constructive and are pleased with how the manuscript was handled editorially. We look forward to future exciting submissions from your lab.

Sincerely,
